# Nitrogen limitation reveals large reserves in metabolic and translational capacities of yeast

Rosemary Yu [1,2], Kate Campbell [1,2], Rui Pereira [1,2], Johan Björkeroth[1,2], Qi Qi [1,2], Egor Vorontsov[3], Carina Sihlbom[3] & Jens Nielsen [1,2,4,5 ✉]

Cells maintain reserves in their metabolic and translational capacities as a strategy to quickly respond to changing environments. Here we quantify these reserves by stepwise reducing nitrogen availability in yeast steady-state chemostat cultures, imposing severe restrictions on total cellular protein and transcript content. Combining multi-omics analysis with metabolic modeling, we find that seven metabolic superpathways maintain >50% metabolic capacity in reserve, with glucose metabolism maintaining >80% reserve capacity. Cells maintain >50% reserve in translational capacity for 2490 out of 3361 expressed genes (74%), with a disproportionately large reserve dedicated to translating metabolic proteins. Finally, ribosome reserves contain up to 30% sub-stoichiometric ribosomal proteins, with activation of reserve translational capacity associated with selective upregulation of 17 ribosomal proteins. Together, our dataset provides a quantitative link between yeast physiology and cellular economics, which could be leveraged in future cell engineering through targeted proteome streamlining.

[1] Department of Biology and Biological Engineering, Chalmers University of Technology, SE-412 96, Gothenburg, Sweden. [2] Novo Nordisk Foundation Center for Biosustainability, Chalmers University of Technology, SE-412 96, Gothenburg, Sweden. [3] Proteomics Core Facility, Sahlgrenska Academy, University of Gothenburg, SE-413 90, Gothenburg, Sweden. [4] Novo Nordisk Foundation Center for Biosustainability, Technical University of Denmark, DK-2800 Kgs, Lyngby, Denmark. [5] BioInnovation Institute, Ole Måløes Vej 3, DK-2200, Copenhagen N, Denmark. ✉email: nielsenj@chalmers.se

Many components of the cellular biosynthetic machinery are not used at full capacity. This enables cells to rapidly respond to changing growth environments at the expense of a reduced steady-state growth rate due to constraints on resource allocation[1,2]. Recent studies in *Escherichia coli*[1,3] and yeast[4,5] have indicated that such strategic reserves are kept for both metabolic and translational capacities of the cell. However, to date, neither of these reserves have been systematically quantified. The ribosome reserve (the fraction of non-translating ribosomes) has been estimated to be 8% of the yeast proteome[4]; however, this does not reflect the reserve for total translational capacity, as gene-specific translation efficiency can be modulated when environmental conditions change, such as upon exposure to stress[6]. Precise measurements of these reserves would improve our understanding of the cell economy and permit biosynthetic processes to be streamlined for synthetic biology applications and/or recombinant protein production efforts. In this study, we have therefore quantified the reserves of the metabolic and translational capacities of the eukaryal model organism *Saccharomyces cerevisiae*.

By stepwise reducing the nitrogen content in the growth medium of yeast steady-state chemostat cultures growing at a fixed dilution rate of $0.2\,h^{-1}$, we have induced global reductions in transcript and protein content in cell biomass until both levels plateaued despite applying further nitrogen restriction, indicating that the minimum transcript and protein content required by the cell is reached. This has revealed that 75% of the total transcriptome and 50% of the proteome are produced in excess of what is necessary to maintain growth. Absolute quantification of the transcriptome and proteome, combined with metabolic flux estimations based on genome-scale modeling, has shown that cells maintain large and unequally distributed reserves in their functional metabolic and translational capacities. Integrated analysis of these data has allowed us to identify regulatory hubs in protein synthesis and metabolism that could subsequently be investigated for their impact on synthetic biology and metabolic engineering outcomes.

## Results

**Transcriptome and proteome allocation are well correlated**. We measured the transcriptome and proteome concentrations (mmol gDW$^{-1}$) of *S. cerevisiae* in four steady-state chemostat conditions in biological duplicates (Supplementary Fig. 1a-c). The growth media were designed to match increasing C/N ratios, by decreasing the concentration of the nitrogen source, ammonium, while maintaining constant glucose concentration (Supplementary Fig. 1a). All cultures were maintained at the same dilution rate of $0.2\,h^{-1}$. Absolute concentrations of 5584 transcripts were obtained by calibrating RNA sequencing (RNAseq) data with a standard curve of the abundance of 31 transcripts covering the entire dynamic expression range (Supplementary Fig. 1d). For quantification of the proteome, we first determined the absolute concentration of proteins in a reference sample by mass spectrometry (MS), using intensity-based absolute quantification (iBAQ)[7] with Proteomics Dynamic Range Standard (UPS2) as the internal standard (Supplementary Fig. 1e). The absolute concentrations of 3483 proteins in each sample were then determined by Tandem Mass Tag (TMT)-based MS[8], using the pooled sample as the internal reference, resulting in an integrated multi-omics dataset that contained 3368 transcript–protein pairs across four steady-state conditions (Supplementary Data 1). Consistent with previous studies[6,9], protein abundance spans a much larger range than transcript abundance. The Pearson's correlation coefficient ($r$) between protein and transcript abundance for each condition ranged between 0.39 and 0.46 (Fig. 1a

and Supplementary Fig. 2a–c), consistent with many prior studies[6,7,9].

As resource allocation plays a key role in controlling the physiological parameters of a cell, we calculated the molar percentage of the transcriptome and proteome allocated to 99 biological processes, as annotated by Yeast Gene Ontology (GO)-slim terms[10] (Supplementary Data 2). Allocation of the transcriptome and proteome to GO-slim terms were well correlated, with $r = 0.82$–$0.87$ (Fig. 1b and Supplementary Fig. 2d-f) for each condition. A similar analysis, assigning genes to 13 groups based on their physiological functions[4], showed similar results ($r = 0.96$–$0.99$; Supplementary Fig. 3). The number of genes in each GO-slim term showed higher correlation with the transcriptome allocation ($r = 0.73$), but lower correlation with proteome allocation ($r = 0.35$; Supplementary Fig. 4a, b). After normalizing to the number of genes in each process, transcriptome allocation to each process had a narrow distribution range, with >95% being within one order of magnitude, whereas the normalized proteome allocation distributed more broadly (Supplementary Fig. 4c). The correlation between transcriptome and proteome allocation after normalizing to the number of genes in each process remained high, with $r = 0.89$ (Supplementary Fig. 4c). This indicates that a small allocation disparity between different processes at the transcriptome level would be amplified at the proteome level. Indeed, the slope from regressing proteome allocation on transcriptome allocation in the log-log scale is greater than one (Fig. 1b and Supplementary Fig. 2d-f), indicating that transcriptional regulation signals are amplified by protein translation[11].

To further examine the distribution of the transcriptome and proteome to different processes, we tested for enrichment of GO-slim terms in 200-gene sliding windows of transcript and protein abundance[9] (Supplementary Fig. 5). We then summarized the GO-slim terms into six functional categories (Supplementary Fig. 5), representing a high-level organizational map of the transcriptome and proteome in steady-state cells (Fig. 1c-d). As a whole, the transcriptome and proteome were organized similarly, consistent with the high correlation in their respective allocation (Fig. 1b). Most of both resources are dedicated to metabolism and translation/protein processing, and little are used for DNA maintenance, stress response, and cell cycle/organelle-related processes (Fig. 1c, d).

We then examined the proteome and transcriptome allocation to GO-slim terms within each of the six functional categories, for a more detailed breakdown of the proteome–transcriptome relationship. We found that four out of six categories agree with the overall high correlation between transcriptome and proteome allocation ($r > 0.85$), whereas two categories showed low correlation: cell cycle/organelle-related processes and stress response (Supplementary Data 3). This is consistent with previous indications that protein degradation plays particularly large roles in controlling the expression of genes participating in these functions[7,12]. The slopes for each category were also calculated, showing clearly that processes with high resource allocation (metabolism, translation, and transcription) have larger slopes than those with low resource allocation (cell cycle, stress response, and DNA maintenance) (Supplementary Data 3), confirming that differential transcriptome allocation to different processes is amplified at the proteome level.

**Transcriptome and proteome reserves for cellular processes**. The typical elemental composition of yeast dry biomass is ~49% carbon and 9% nitrogen[13] (which likely changes depending on the growth condition), representing a C/N ratio of 5.4. In typical carbon-limited chemostat cultures, nitrogen is provided in excess with C/N ratio of 3–4 in the growth medium[6,14]. Reducing the

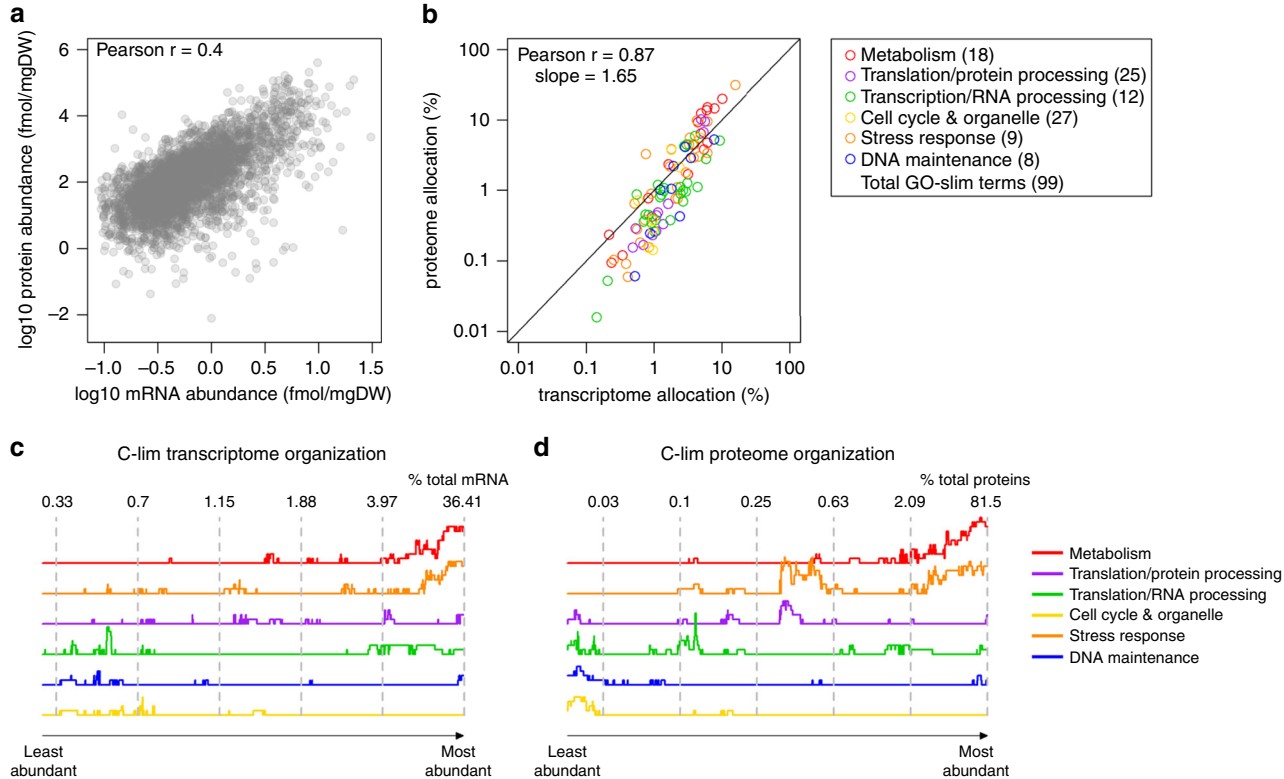

**Fig. 1 Transcriptome and proteome allocation are better correlated at the process level than the gene level. a** Absolute mRNA and protein abundances were quantified in yeast chemostat cultures at a dilution rate $D = 0.2\,h^{-1}$. Mean values of biological duplicates for the carbon-limited condition are shown. See Supplementary Fig. 2D-F for other conditions. **b** Transcriptome and proteome allocations to 99 GO-slim processes under carbon-limited condition were calculated as a % ($mol\,mol^{-1}$) of total transcripts and proteins. GO-slim processes are color-coded by functional category. The number of GO-slim terms curated in each functional category is shown. See Supplementary Data 2 for number of genes in each GO-slim term. **c** Enrichment of GO-slim terms in 200-gene sliding windows of relative transcript abundance was analyzed by Fisher's exact test. Y-axis represents relative # of GO-slim terms belonging in each functional category with FDR-adjusted $p_{Fisher} < 0.05$. Numbers indicate the relative abundance of transcripts (% $mol\,mol^{-1}$ of total transcripts) in the 200-gene window at the dashed lines. **d** As in **c** for proteome organization.

nitrogen content in the growth medium to a C/N ratio of 50–115 (Supplementary Fig. 1a) severely limits the supply of nitrogen needed for the biosynthesis of amino acids and nucleotides, leading to total protein and mRNA in the cell declining to 50% and 25% of the amount measured in C-limited cultures, respectively (Fig. 2a). Further reductions in nitrogen availability reduced the steady-state biomass concentration but did not further decrease the RNA and protein content per gram of dry weight, suggesting these levels to be the minimum required for cell growth at a constant dilution rate of $0.2\,h^{-1}$ (Fig. 2a). Under these nitrogen-limiting conditions, therefore, we consider the transcriptome and proteome allocation of the cell to be fully economized. Remarkably, the allocation of the proteome and transcriptome to different cellular processes did not change between C-limited and N-limited cultures for nearly all processes (97 out of 99 GO-slim terms; $-1 < log2 < 1$; Fig. 2b-c, Supplementary Fig. 6, and Supplementary Data 2). A similar analysis, assigning genes to 13 groups based on their physiological functions[4], showed similar results (Supplementary Fig. 7). In other words, in each process 50% of proteins and 75% of mRNA are maintained as reserves under carbon-limited conditions. These results show that, although resource allocation is known to vary with cell growth rate[2,4], they are insensitive to changing growth environments. As an example, we see that for the core central carbon metabolism (CCM) pathways, the total proteomic allocation (red symbols; Fig. 2d-f) remained constant for all conditions, despite dramatic differences in the total protein content (Fig. 2a) and distribution of carbon flux (Supplementary Data 4).

Within each pathway, although the total proteome allocation remained largely constant, the abundance of individual enzymes can be adjusted to suit the metabolic need of the cell. For example, Adh2 was downregulated by up to 131-fold as nitrogen content is reduced (Fig. 2e), likely due to glucose repression as residual glucose concentration rises[15]. Despite this large differential expression in Adh2, proteome and transcriptome allocation to the fermentation pathway remained largely constant (Fig. 2e and Supplementary Fig. 8). We then examined the proportion of genes in each process that were adjusted by more than twofold in protein and transcript allocation between N-limited and C-limited cultures ($-1 < log2 < 1$; Supplementary Fig. 9 and Supplementary Data 5). We found that 9 of the top 10 processes with the highest proportion of differentially allocated individual proteins were related to metabolism (Supplementary Fig. 9 and Supplementary Data 5), indicating that resource allocation to individual metabolic enzymes are subject to a higher degree of fine tuning compared with proteins in non-metabolic processes.

To validate the finding that resource allocation is constant to the cell growth rate, we mined the multi-omics dataset from a previous study[6], where 1625 transcript–protein pairs were measured in 10 *S. cerevisiae* chemostat cultures at a constant dilution rate of $0.1\,h^{-1}$. Transcriptome and proteome allocation to the 99 GO-slim terms in this dataset were also constant across all conditions ($-1 < log2 < 1$; Supplementary Fig. 10). This confirms that we had uncoupled the effect of growth rate and the effect of growth environment on resource allocation.

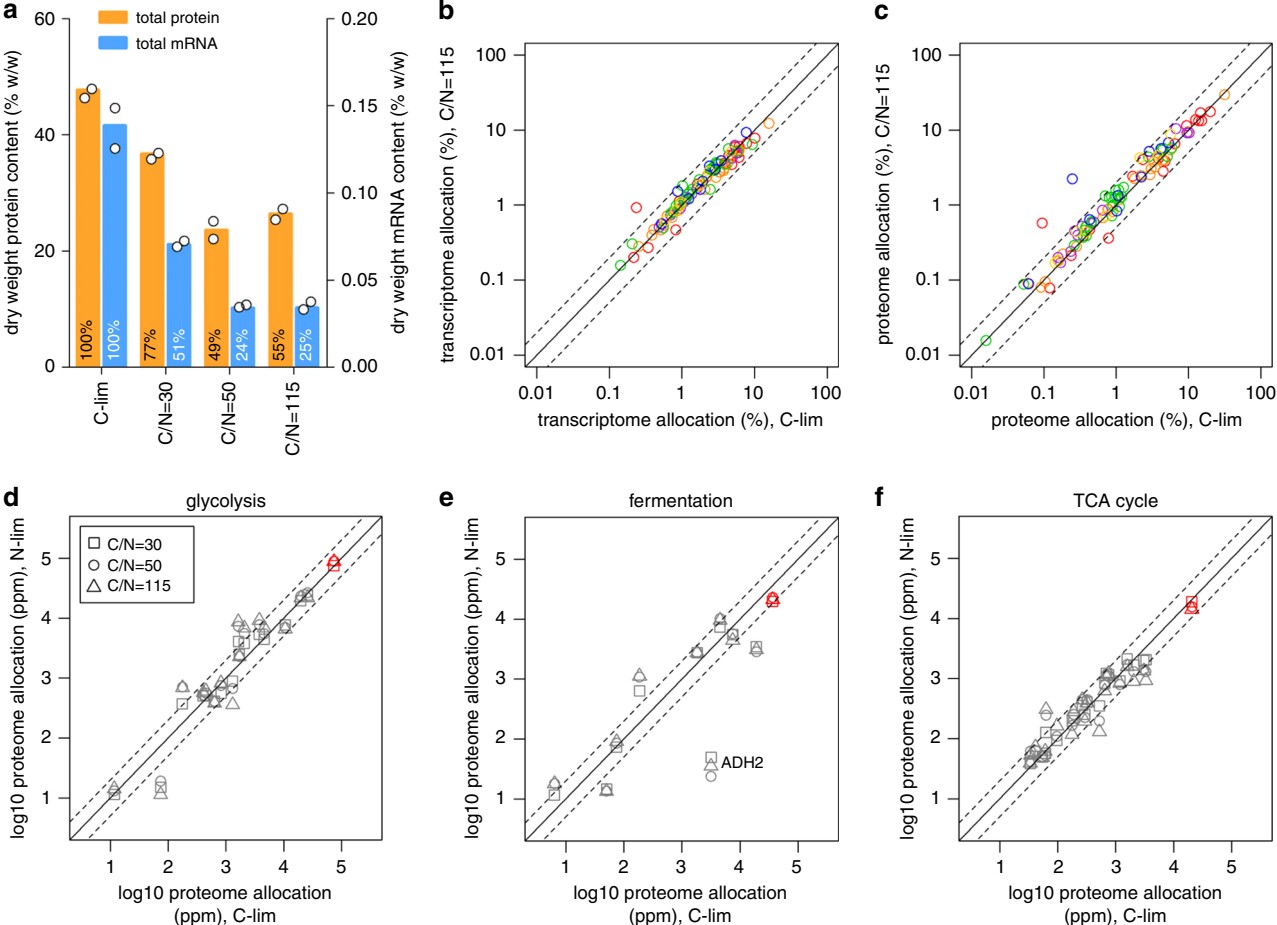

**Fig. 2 Transcriptome and proteome reserves for different cellular processes. a** Total protein and mRNA content of the biomass under each culture condition. Colored bars are mean of biological duplicates. **b** Transcriptome allocation (% mol mol$^{-1}$) to 99 GO-slim terms were compared between N-limited and C-limited cultures. GO-slim terms are color-coded by functional category as in Fig. 1b. Dashed lines are twofold increase and decrease from $y = x$, solid line. Mean values for carbon-limited and C/N = 115 conditions are shown. The red outlier represents the "amino acid transport" GO-slim term. **c** As in **b** for proteome allocation. The red and blue outliers represent "amino acid transport" and "response to starvation" GO-slim terms, respectively. **d** Proteome allocation to each enzyme in glycolysis was compared between N-limited (C/N = 30, 50, and 115) and C-limited cultures. Total proteome allocation to all enzymes in glycolysis is shown in red. Dashed lines are twofold increase/decrease from $y = x$, solid line. ppm, parts per million (mol mol$^{-1}$). **e** As in **d** for proteome allocation to enzymes in fermentation. Total proteome allocation to all enzymes in fermentation is shown in red. Dashed lines are twofold increase/decrease from $y = x$, solid line. **f** As in **d** for proteome allocation to enzymes in the TCA cycle. Total proteome allocation to all enzymes in the TCA cycle is shown in red. Dashed lines are twofold increase/decrease from $y = x$, solid line.

These combined results suggest that growth rate modulates total resource allocation for different cellular processes, whereas changing the growth environment modulates the ratio of expression of different genes within each process, particularly for processes related to metabolism.

**Metabolic superpathways maintain large reserve capacities.** Previous studies in *E. coli*[16,17] have indicated that cells maintain large reserves in their metabolic capacity to manage fluctuating environments[1]. Quantification of these reserves from such studies, however, can be difficult to obtain. Conceptually, one could estimate these reserves by extrapolating from linear/polynomial correlations between enzyme abundance and the specific growth rate $\mu$, as $\mu \to 0$. However, the abundance of enzymes in CCM is negatively correlated with $\mu$[16,17], clearly indicating that the size of the metabolic reserve can vary with growth rate. Here we estimated for the first time the cellular enzyme reserve without the confounding factor of a changing growth rate, by estimating the relative enzyme usage in each of our experimental conditions using enzyme-constrained (ec) genome-scale metabolic modeling

(GEM)[18]. GEM simulations are based on the concept of flux balance analysis (FBA)[19], which balances the influx and efflux around each metabolite in the network, and is widely applicable for metabolic modeling of steady-state conditions such as chemostat cultures[19,20]. It should be noted, however, that GEM simulations do not take intracellular metabolite concentrations into consideration (as, in any given steady state, metabolite concentrations do not change), and therefore the influence of different metabolite concentrations between different steady states[21] are not considered.

In the ecGEM of yeast, ecYeast8.1, a pseudo-metabolite is used to represent the total enzyme pool ($P_{met}$)[18]. Constraining $P_{met}$ limits the total amount of enzyme that the model can use to simulate yeast metabolism[18]. In our dataset, a constant 52–55% (g g$^{-1}$) of the measured proteome were metabolic enzymes, therefore we used these measurements to constrain the upper bound of $P_{met}$ in each condition. We then applied additional constraints based on measured metabolite exchange fluxes, biomass composition, and the dilution rate (Supplementary Data 6). Flux variability analysis (FVA)[22] using these condition-

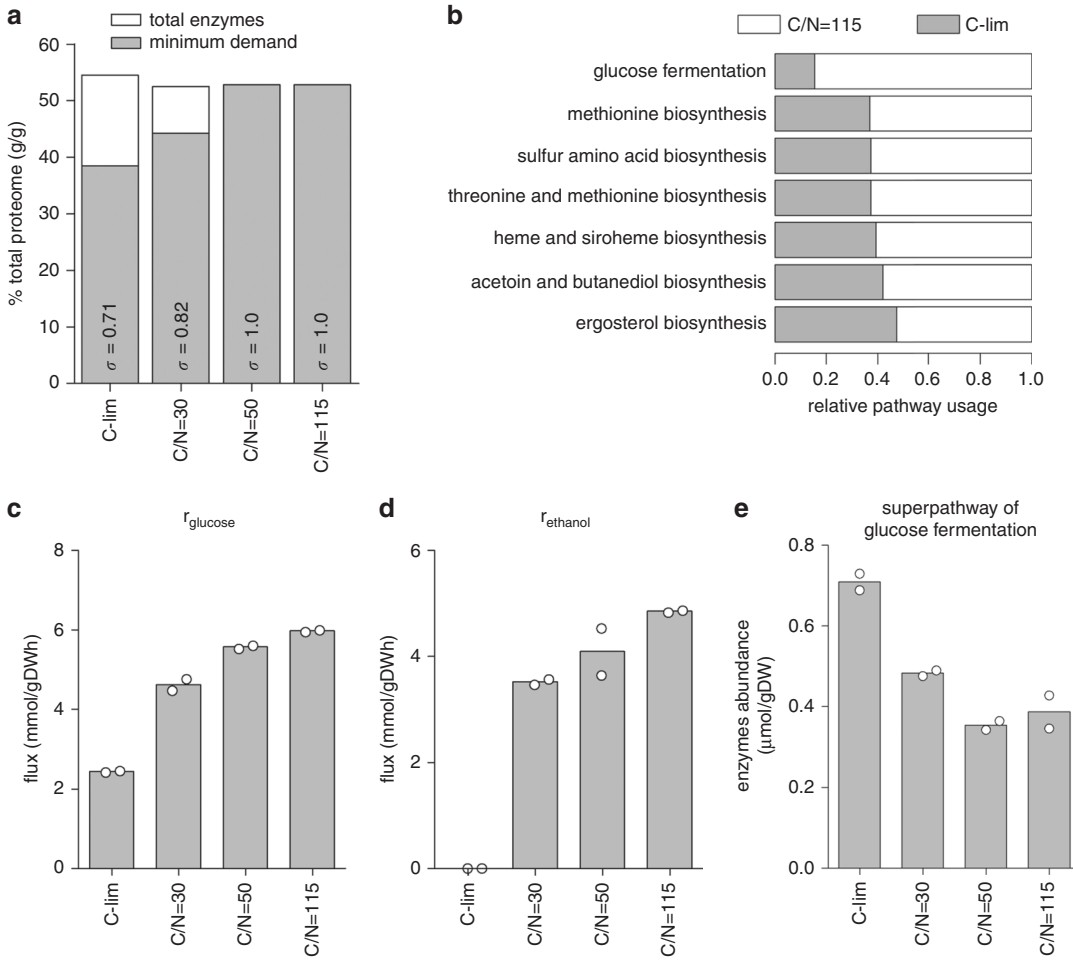

**Fig. 3 Metabolic superpathways maintain large reserve capacities in carbon-limited cultures. a** Total enzyme abundance is compared with the minimum enzyme demand calculated in silico by ecYeast8.1 for each condition. The enzyme saturation coefficient $\sigma$ is shown (see Methods). **b** Enzyme usage for selected yeast metabolic superpathways. Flux balance analysis (FBA) with random sampling was implemented in ecYeast8.1 to estimate the optimal enzyme requirement for each enzyme (see Methods). Pathway usage was computed as the sum of simulated enzyme requirement divided by the sum of measured enzyme abundance in the pathway. Pathways with relative usage <50% between C-limited and C/N = 115 cultures are shown. **c** Glucose consumption rate ($r_{glucose}$) was measured under different culture conditions. Bars are mean of biological duplicates. **d** Ethanol production rate ($r_{ethanol}$) was measured under different culture conditions. Bars are mean of biological duplicates. **e** Absolute abundance of enzymes in the superpathway of glucose fermentation was calculated. Bars are mean of biological duplicates.

specific models showed large reductions in the degrees of freedom on the model simulations where nitrogen is reduced, pointing to increased enzyme usage and a reduced solution space. We then used these models to estimate the theoretical minimum enzyme demand in silico. This was done by setting the objective function of the model simulations to minimize $P_{met}$. This allowed the calculation of the enzyme saturation coefficient ($\sigma$)[18], which is functionally equivalent to the fraction of $P_{met}$ that corresponds to the in-silico minimum enzyme demand (Fig. 3a). From this, we estimated total enzyme reserve to be 29% of $P_{met}$ or 14% of the total proteome at C-limited growth at 0.2 h$^{-1}$ (Fig. 3a).

As nitrogen was reduced to reach a C/N ratio of 50–115, the minimum enzymatic demand calculated in silico equaled the amount of enzymes measured in vivo (Fig. 3a), confirming that enzyme usage was maximized under these conditions. We next quantified the reserve capacity for all metabolic superpathways, as defined by YeastPathways[10], by comparing their relative usage in C-limited and C/N = 115 cultures. We constrained ecYeast8.1 with an upper bound for the exchange reaction of the enzyme pool pseudo-metabolite equal to $P_{met} \cdot \sigma$ and implemented FBA with random sampling[23] to compute the optimal solution for

each culture condition. Pathway usage was calculated as the sum of the model-calculated enzyme requirement divided by the sum of experimentally measured abundance values of all enzymes in each metabolic superpathway (Supplementary Data 7). We found that 7 out of 23 superpathways maintained >50% metabolic capacity as reserves when cells were growing in C-limited culture (Fig. 3b), whereas the remaining superpathways had very little metabolic capacity left as reserve.

We benchmarked these results using the relative usage of the superpathway of glucose fermentation. At C/N = 115, the glucose consumption rate was measured to be 6.0 mmol gDW$^{-1}$ h$^{-1}$ (Fig. 3c and Supplementary Fig. 1b), a 2.5-fold increase compared with the 2.4 mmol gDW$^{-1}$ h$^{-1}$ glucose consumption in C-limited cultures. This is accompanied by a switch from respiratory metabolism (producing only $CO_2$) in C-limited cultures, to respiro-fermentative metabolism (producing $CO_2$, ethanol, and various organic compounds) in N-limited cultures (Fig. 3d and Supplementary Data 4). Meanwhile, there is a 50% reduction in enzyme abundance (Fig. 3e), indicating a fivefold increase in pathway usage. In other words, in C-limited cultures, only 20% of the total metabolic capacity of this superpathway was used, a

result which is in good agreement with the computationally estimated pathway usage (Fig. 3b). Maintenance of these large reserves also fits well with previous observations that metabolic fluxes in the CCM are predominantly controlled posttranslationally, in both bacteria[17,24] and yeast[6].

In Supplementary Data 8, we provide the flux estimated by ecYeast8.1 for each enzyme in the 23 superpathways. For the superpathways of methionine, sulfur amino acid, and threonine and methionine biosynthesis (Fig. 3b), Met17 alone accounted for nearly 50% of the reserves (Supplementary Fig. 11). For most other superpathways, however, there is not a single enzyme that makes up such a large fraction of the reserves. Of interest is whether rate-limiting enzymes (RLEs)[25] showed a greater change when nitrogen content in the growth media was stepwise reduced. We found that RLEs have a very slight, although significant ($p < 0.05$), larger proteome allocations compared with non-RLEs (Supplementary Fig. 12).

We also examined whether expression of isozymes (Supplementary Fig. 13a-b) represents a major form of metabolic reserve. In our dataset, 65 of 238 isozymes (27%) showed differential allocation between C-limited and N-limited cultures (Supplementary Fig. 13a-b and Supplementary Data 9). However, we noted that "switching" between different isozymes (e.g., Supplementary Fig. 13c-e) rarely occurs: e.g., although the relative expression of Gdh3 changed by 3.6-fold, it remained the minor isozyme of glutamate dehydrogenase, with Gdh1 constituting >99% of this enzyme across all conditions (Supplementary Data 9).

**Translational reserves are preferentially used for enzymes.** In addition to metabolic reserves, cells also maintain reserves in translational capacity[1,4,5,26]. In our dataset, as the total protein-to-mRNA ratio was doubled under N-limitation (Fig. 2a), this places the overall translational reserve at ∼50% capacity in C-limited cultures. As both the transcriptome and proteome allocation for different cellular processes maintained a similar % of reserves (Fig. 2b-c), we next investigated whether this is true for cellular reserves in translational capacity as well.

Using protein and mRNA abundances from this dataset and mining protein turnover data from Lahtvee et al.[6], we calculated protein synthesis efficiency $k_{sP}$ (protein mRNA$^{-1}$ h$^{-1}$) for each protein (Supplementary Fig. 14a-b and Supplementary Data 10)[6,7]. We found that $k_{sP}$ correlated well with protein abundance, with $r = 0.70$–$0.72$ (Supplementary Fig. 14c-f), indicating that protein abundance is determined by protein translation rate to a higher degree than by mRNA abundance (Fig. 1a and Supplementary Fig. 2a-c). With stepwise reduction of nitrogen content in the growth media, $k_{sP}$ globally increased (Fig. 4a), indicating that reserve translational capacities were placed into usage. Surprisingly, the relative increase in $k_{sP}$ in each condition was not the same for all genes (Supplementary Data 10). To study this response, we grouped genes based on the step of nitrogen reduction at which $k_{sP}$ was increased by >2-fold (Supplementary Fig. 15a-c), which splits a total of 3361 genes into 4 groups of similar sizes (Fig. 4b and Supplementary Data 10). When nitrogen availability was at its lowest, at C/N = 115, genes in group 1 had a median $k_{sP}$ increase of 5.6-fold (Fig. 4b), i.e., an 82% reserve translational capacity under typical C-limited growth. Remarkably, this group of genes were enriched exclusively in processes related to metabolism (Fig. 4c, group 1). How metabolic genes can be enriched in this group, while the total proteomic and transcriptomic allocation to metabolic processes remained fixed (Fig. 2b-c), can be explained by the observation that abundance of metabolic enzymes had a higher propensity to be internally adjusted within a given metabolic

pathway (Supplementary Fig. 9 and Supplementary Data 5). Taken together, these results show that the vast majority of cellular reserves are dedicated to metabolism.

At the final step of nitrogen reduction in the growth medium, a total of 2490 out of 3361 genes (74%) exhibited a > 2-fold $k_{sP}$ increase, demonstrating >50% reserve in their translational capacities when growing in C-limited conditions (Fig. 4b and Supplementary Fig. 15a-c). The remaining 26% of genes showed little to no modulation of $k_{sP}$ and were enriched in translation/protein processing-related GO-slim terms (Fig. 4c, group 4). This indicates that components of the translational machinery were themselves being translated at a maximum capacity in C-limited cultures, and changes in the abundance of these proteins are regulated transcriptionally. Of note, the differential use of translational reserves was neither correlated with mRNA abundance nor with changes in mRNA abundance between conditions (Supplementary Fig. 16). Finally, we observed no significant enrichment ($p_{Fisher} > 0.01$) in any of the four groups herein for essential genes (Supplementary Fig. 15d)[10], indicating that the unequal distribution of these reserves is not a critical feature for survival, but likely arose by conveying a selective advantage and increased fitness.

**Ribosome reserves contain sub-stoichiometric RPs.** The observation that cells preferentially maintain reserve translational capacities for some processes (metabolism) but not others (components of the translational machinery) is interesting, considering that functionally distinct sub-pools of mRNA are known to be translated by subsets of ribosomes with distinct ribosomal protein (RP) stoichiometry[27–29]. RP stoichiometry has also been shown to depend on the balance between the economics of protein production and cell growth[30]. To investigate this further, we examined RP stoichiometry in our proteomics dataset for C-limited cultures and found it to span >2 orders of magnitude, even when their paralogs were summed (Supplementary Fig. 17). This large variation could not be accounted for by differences in specific RP characteristics[6,31] or parameters of the MS (Supplementary Fig. 18). Interestingly, most deviations from mean RP abundance and rRNA abundance were sub-stoichiometric RPs (Fig. 5a and Supplementary Fig. 17), indicating that a substantial number of ribosomes in the cell likely contain a sub-stoichiometric composition of available RPs.

To validate this finding, we performed targeted quantitative proteomics for 26 RPs by TMT-based MS, using 49 proteotypic peptides that were chemically synthesized at known quantities as standards (Supplementary Data 11). The synthetic peptide standards and the pooled reference sample were multiplexed at eight different ratios to ensure that MS intensity ratios cover the dynamic range of TMT[32]. This analysis confirmed that iBAQ-based quantification is robust to the order of magnitude, for 23 out of 26 (88%) RPs (Supplementary Fig. 19a and Supplementary Data 11). This analysis further indicated that any posttranslational modifications on these peptides did not significantly interfere with the absolute quantification by MS, as the synthetic peptide standards are not modified. Thus, we confirmed that the abundance of RP subunits in a cell is markedly different from the typically assumed 1 : 1 stoichiometry. Out of the 76 RP subunits detected in our dataset, 54 (70%) were expressed in the order of $10^5$ molecules per cell (mean = $3 \times 10^5$ molecules per cell), in line with classic estimates of ribosome content[33], whereas 22 (30%) RP subunits were sub-stoichiometric by up to an order of magnitude (Supplementary Fig. 17 and Supplementary Data 12).

From this data, a simple model of ribosome complex diversity and abundance emerges (Fig. 5b, c and Supplementary Fig. 19b, c).

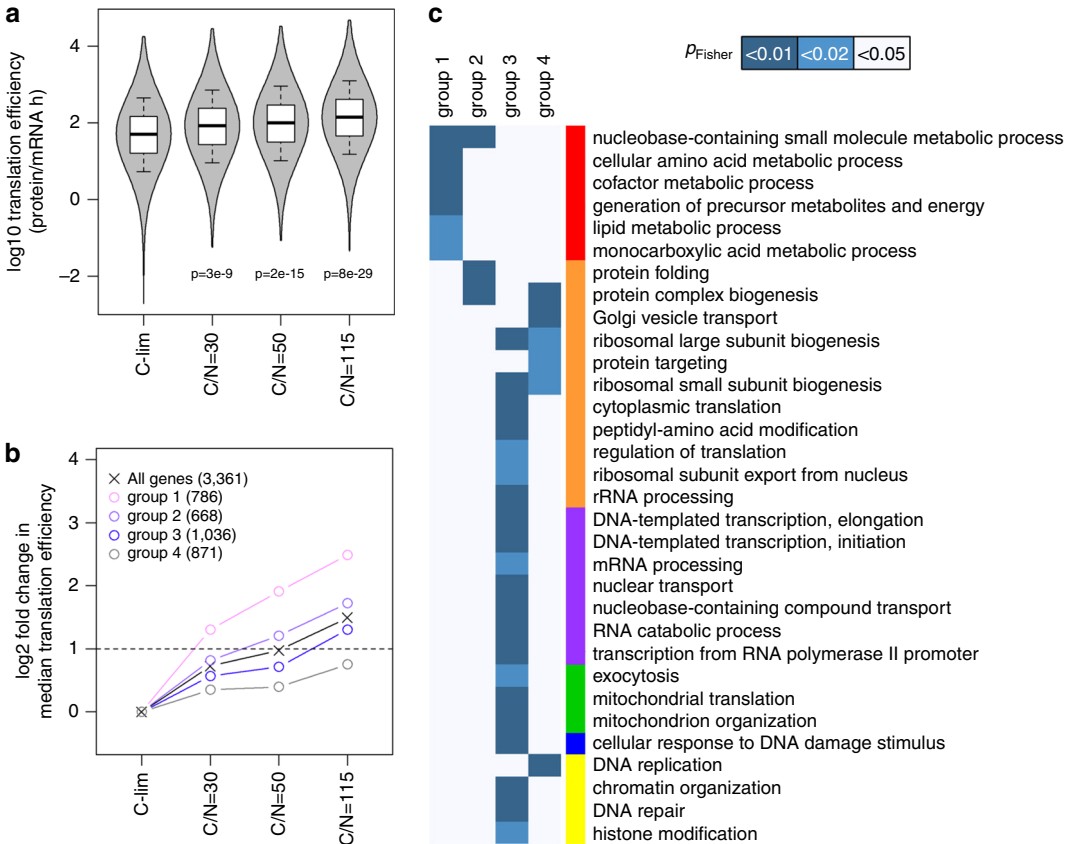

**Fig. 4 Reserves of translational capacity are preferentially used to translate metabolic proteins. a** Gene-specific translation efficiency was calculated in all growth conditions. The *p*-values indicated are from two-sided Student's *t*-tests. Center line, median; box limits, upper and lower quartiles; whiskers, 1.5× interquartile range. **b** Genes were grouped based on the step of nitrogen reduction at which their translation efficiency was increased by log2 > 1. Data points reflect the median translation efficiency of each group, with the # of genes belonging to each group shown in parentheses. Horizontal dashed line indicates twofold increase from C-limited cultures. **c** Enrichment of GO-slim terms in each gene group as defined in **b** was analyzed by two-sided Fisher's exact test. GO-slim terms are color-coded by functional category as in Fig. 1b. GO-slim terms with FDR-adjusted $p_{Fisher} < 0.05$ in at least one gene group are shown.

Each complexed ribosome likely contains all or most of the 54 "core" RP subunits that are expressed at high abundance and only a subset of the 22 RP subunits that are expressed an order of magnitude lower (Fig. 5b and Supplementary Fig. 19b). Thus, the total number of ribosomes that can be complexed with *n* RP subunits decreases with increasing *n* when *n* > 54 (Fig. 5c and Supplementary Fig. 19c). The diversity of ribosome subunit composition varies quadratically with *n*, with peak diversity occurring with complexes of *n* = 64–65 RP subunits, where >$10^6$ combinations are theoretically possible (Fig. 5c and Supplementary Fig. 19c). Notably, for complexes of *n* = 58–71, the number of possible RP subunit combinations exceeds the actual number of ribosomes that can be built with *n* RP subunits (Fig. 5c). This means that it is possible for these ribosomes to re-assemble into completely new RP subunit compositions if needed, pointing to a dynamic pool of ribosome reserves that can quickly respond to changing growth environments. In addition, the abundance of mitoribosomal protein (MRP) subunits also spanned >2 orders of magnitude, with ~70% of MRPs being >10-fold more abundant than the other 30%, similar to the distribution of cytoplasmic RPs. In contrast, the subunit stoichiometry of several other multi-protein complexes[34] were largely within the same order of magnitude (Supplementary Fig. 20). This suggests that subunit stoichiometry of both cytoplasmic and mitochondrial ribosomes are more flexible and diverse, whereas other protein complexes may require more rigid subunit stoichiometries to fulfill their functions.

As nitrogen is reduced in the growth media, less RPs are detected at sub-stoichiometric levels to rRNA (Fig. 5a), with 17 RP subunits (22 RPs) being selectively upregulated by >2-fold (Fig. 5d and Supplementary Fig. 21a). This upregulated RP pool includes three RP subunits wherein a paralog-specific response to nitrogen reduction was observed (Supplementary Fig. 21d). Interestingly, not all RPs upregulated in the N-limited cultures were sub-stoichiometric in the control C-limited culture and vice versa (Supplementary Fig. 21b, c), implicating multiple levels of control for RP subunit incorporation and ribosomal translation efficiency. Of note, the stoichiometries of several proteasomal subunits were also changed by >2-fold with decreasing nitrogen, but not in any of the catalytic subunits or ATPases[35] (Supplementary Fig. 22). Taken together, these data suggest that cells become more selective in both protein translation and degradation when nitrogen is limiting.

## Discussion

There is growing evidence in both *E. coli*[1,3] and yeast[4] that cells maintain reserve capacities in metabolism and protein translation, trading off maximum exponential growth rate for the ability to respond quickly to changes in their growth environment. Here we quantified for the first time the sizes of reserves of the transcriptome, proteome, metabolic capacity, and translational capacity, without the confounding factor of a changing growth

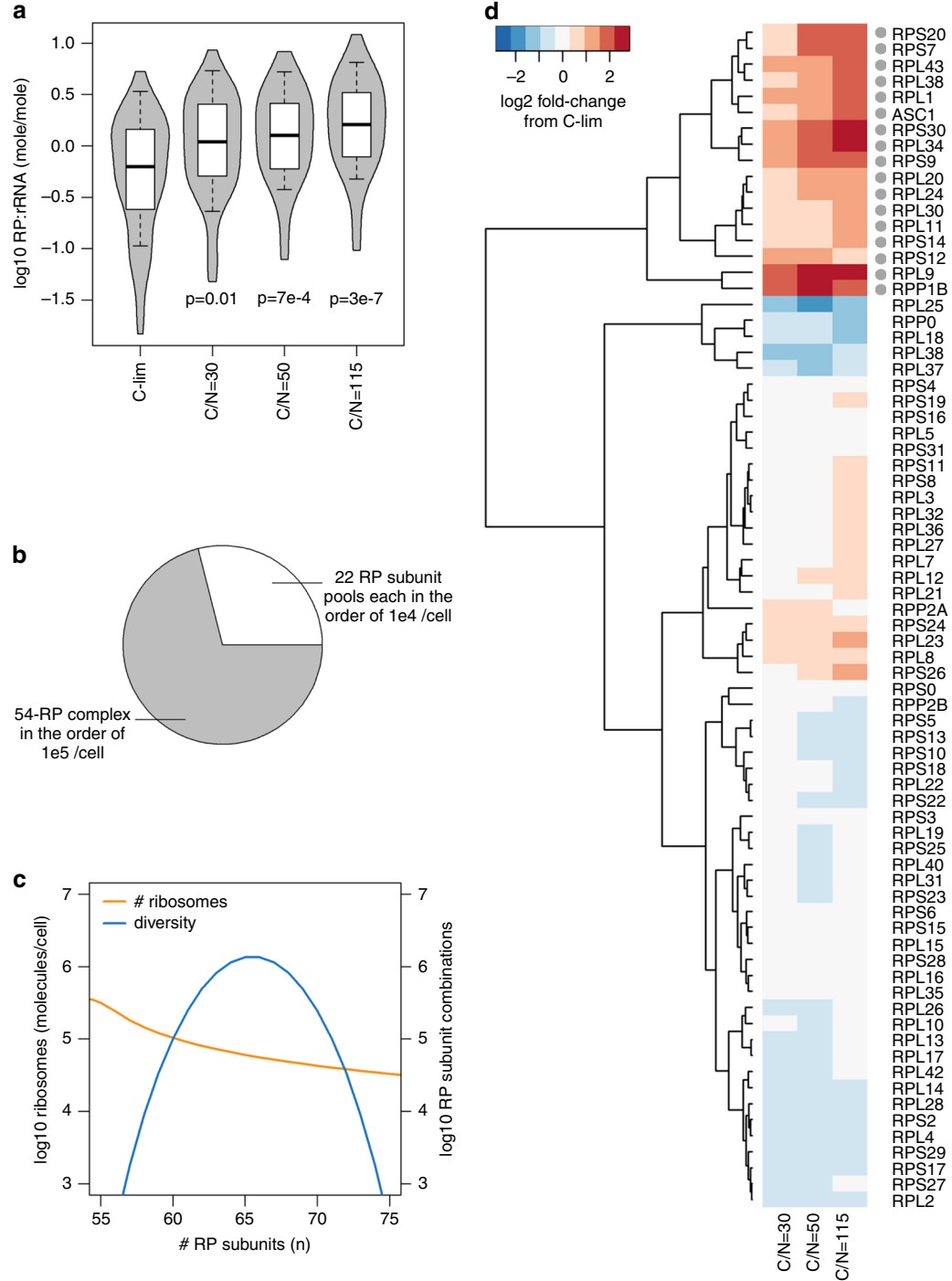

**Fig. 5 Ribosome reserves contain diverse sub-stoichiometric ribosomal proteins (RP). a** RP : rRNA ratio for each RP (abundance of each RP divided by rRNA abundance) was calculated in all growth conditions. The *p*-values indicated are from two-sided Student's *t*-tests. Center line, median; box limits, upper and lower quartiles; whiskers, 1.5× interquartile range. **b** Model of ribosome complex diversity and abundance. Each complex ribosome contains 54 "core" RP subunits that are highly expressed and samples a subset of 22 RP subunits expressed an order of magnitude lower. **c** The total number of ribosomes in the cell decreases, whereas diversity of ribosomes varies quadratically, with the number of RP subunits complexed. **d** Relative abundance of RP subunits in N-limited chemostats compared with C-limited chemostat. RP paralogs were summed and labeled with standard gene name. The 17 RP subunits that were selectively upregulated by >2-fold in N-limited chemostats are indicated with a gray dot.

rate. Overall, 50% of the proteome serves as a reserve, alongside 75% of the transcriptome, 30% of metabolic capacity, and 50% of translational capacity (Supplementary Fig. 23). Our analysis showed that a major part of these reserves is preferentially maintained for metabolic processes. Building on numerous prior studies of systems-level responses to nutrient limitations[21,36,37],

our results highlight the importance of a robust metabolism for cell growth.

Optimal allocation of the cell transcriptome and proteome to different cellular process is integral to maximizing cellular output and fitness. For example, to reach high growth rates, yeast cells allocate more of its proteome to the translation

machinery (from <10% to >30%), while reducing the proteome allocation towards glycolysis (from 10% to <5%) and other CCM pathways[4]. Surprisingly, we found in this work that, in the context of a changing growth environment without a change in growth rate, transcriptome and proteome allocation to nearly all cellular processes remained constant. This allowed us to uncouple the effect of growth rate and the effect of growth environment on cell physiology and, hereby, for the first time, show that growth rate determines the allocation of the proteome and transcriptome to different processes, while growth environment modulates the gene expression ratio within each process.

We also found that several metabolic pathways accommodated for equal (if not increased) flux with decreased enzyme abundance when nitrogen is reduced, clearly indicating that cells maintain large metabolic overcapacities in C-limited conditions. In particular, CCM was found to contain >500% overcapacity in C-limited growth conditions. Of note, our data alone cannot differentiate whether this exists in the form of excess/unused enzymes, or in the form of reduced enzyme catalytic efficiency caused by unfavorable kinetic and thermodynamic constraints. However, two previous studies have independently shown that laboratory evolution of *S. cerevisiae* in continuous C-limited cultures gives rise to evolved strains with reduced glycolytic activities, to as low as 13% that of the parental strains[38,39]. This suggests that glycolytic enzymes are indeed kept in excess of demand in the parental strains, and loss of these excess enzymes provides a selective advantage by reducing the proteomic burden. Moreover, as glucose was the growth-limiting nutrient where cells maintained >500% overcapacity in glucose metabolism, it is unlikely that this is necessary to alleviate metabolic self-inhibition[40].

The exceptionally large CCM reserve capacity raises the question of whether these enzymes are particularly highly expressed simply because glucose is limiting. Indeed, when glucose is more available (which, when glucose is the limiting nutrient, allows for faster growth), CCM reserves are reduced[16,17]. However, such a reduction of CCM reserves with increasing growth rate also occurs when glucose is in excess, including when auxotrophic strains are limited by Leu or Ura, where C and N are both available in excess[37]. Thus, the CCM metabolic reserve is likely a general feature of the cell economy. In the natural setting, although *S. cerevisiae* has been suggested to be a "nomad" species with no specific niche[41], our data indicate that it would be well-adapted to a lifestyle consisting of long periods of C-limitation, e.g., in the soil[42], with occasional bursts of C availability. Yeast cells could therefore have been selected to maintain large reserves in the metabolic capacity for essential pathways such as the CCM, to ensure that these pathways remain unobstructed when the growth environment abruptly changes.

Previous work[43] has shown that in N-limitation, the cellular glycogen content increases from 2.5% to 15–20% of the cell dry weight and trehalose increases from 0.3% to 10–15%. Our data also implicates increased lipid content, as protein and RNA content are reduced in N-limited conditions, with respiratory quotient > 1 suggesting lipogenesis (Supplementary Data 4). We note that proteomic and transcriptomic allocations to the metabolic pathways of glycogen, trehalose, and lipids remained constant between C-limited and N-limited cultures, further supporting that cells maintain large reserves in the metabolic capacity of a variety of biomolecules. Moreover, with these large metabolic reserves in place, transcript-level differences in enzyme expression would be expected to have a low impact on metabolic output, as has been shown in many previous studies[21,36,38].

Our multi-omics dataset also allowed for an in-depth analysis of RP stoichiometry and protein translation efficiency, supporting the idea that a large diversity of ribosomes could be present in the cell and be responsible for the wide range of gene-specific $k_{sP}$[44,45]. Indeed, several landmark studies have shown that ribosomes with or without a specific RP can translate functionally distinct sub-pools of mRNA[27–29], collectively known as the ribosome code[46,47]. We show here with high confidence that 17 RP subunits (22 RPs) are selectively upregulated under nitrogen-limited conditions, expanding the ribosome code and our understanding of the protein translation process as well. These observations combined suggests the intriguing possibility that RP stoichiometry can be modulated to engage reserve translational capacities, and may be optimized to improve translation efficiency for synthetic biology and metabolic engineering applications.

Our data also demonstrated that, when confronted with a decreased supply of mRNA, cells can meet the protein synthesis demand by increasing the gene-specific translation efficiency for 74% of genes. This could be actively regulated, or could arise naturally following a limit in the amount of ribosomes in the cell. For the remaining 26%, enriched for translation/protein-processing functions, protein abundance is predominantly regulated by transcript abundance. This is in line with a recently described model of upshift kinetics during famine-to-feast transition[1], where the increase in growth rate upon nutrient upshift is characterized by first an instantaneous jump, followed by a slow increase to the final growth rate. This first "jump" is related to an immediate boost in protein synthesis by engaging reserve translational capacities[1], which we demonstrated here to preferentially translate metabolic proteins. The de novo synthesis of additional ribosomes and other components of the translational machinery are much slower in comparison, which would give rise to the slow adaptation phase of the upshift kinetics[1]. Thus, in addition to providing quantitative measurements of multiple levels of cellular reserves, our data also provides a framework for modeling complex cellular behavior, an outstanding challenge in systems and synthetic biology.

## Methods

**Culture conditions**. The yeast *S. cerevisiae* CEN.PK 113-7D (MATa, MAL2-8c, SUC2) was used for all experiments. Cells were stored in aliquoted glycerol stocks at −80 °C. Chemostat experiments were carried out under carbon or nitrogen-limited conditions on minimal mineral medium at a constant specific growth rate of 0.2 h⁻¹, at 30 °C, pH 5, working volume 0.5 L, aeration 1 v.v.m., pO₂ > 30%, agitation speed 800 r.p.m. Chemostat experiments were carried out in DASGIP 1 L bioreactors (Jülich, Germany) equipped with off-gas analysis, pH, temperature, and dissolved oxygen sensors. Chemostat medium contained glucose and (NH₄)₂SO₄ as indicated in Supplementary Fig. 1a, as well as the following: KH₂PO₄, 3 g L⁻¹; MgSO₄·7H₂O, 0.5 g L⁻¹; trace metals solution, 1 ml L⁻¹; vitamin solution, 1 ml L⁻¹; antifoam, 0.1 ml L⁻¹. The trace metal solution contained the following: EDTA (sodium salt), 15.0 g L⁻¹; ZnSO₄·7H₂O, 4.5 g L⁻¹; MnCl₂·2H₂O, 0.84 g L⁻¹; CoCl₂·6H₂O, 0.3 g L⁻¹; CuSO₄·5H₂O, 0.3 g L⁻¹; Na₂MoO₄·2H₂O, 0.4 g L⁻¹; CaCl₂·2H₂O, 4.5 g L⁻¹; FeSO₄·7H₂O, 3.0 g L⁻¹; H₃BO₃, 1.0 g L⁻¹; and KI, 0.10 g L⁻¹. The vitamin solution contained the following: biotin, 0.05 g L⁻¹; *p*-amino benzoic acid, 0.2 g L⁻¹; nicotinic acid, 1 g L⁻¹; Ca-pantothenate, 1 g L⁻¹; pyridoxine-HCl, 1 g L⁻¹; thiamine-HCl, 1 g L⁻¹ and myo-inositol, 25 g L⁻¹.

**Sampling from bioreactor**. The dead volume was collected with a syringe and discarded. For transcriptome sampling, biomass was collected from the reactor with a syringe and injected into chilled 50 ml Falcon tubes filled with 35 mL crushed ice. Samples were centrifuged for 4 min at 3000 × *g* at 4 °C; cell pellets were washed once with 1 mL of chilled water, transferred into Eppendorf tubes, flash frozen in liquid nitrogen, and stored at −80 °C until analysis. For proteome sampling, biomass was collected from the reactor with a syringe and injected into 50 ml Falcon tubes chilled on ice. Samples were centrifuged for 4 min at 3000 × *g* at 4 °C; cell pellets were washed once with 20 ml of chilled dH₂O, washed again with 1 ml of chilled water, transferred into Eppendorf tubes, flash frozen in liquid nitrogen, and stored at −80 °C until analysis. Biomass determination was done by filtration of the culture broth on pre-weighed filter paper, drying in a microwave at 360 W for 20 min, and desiccating in a desiccator for >3 days. Exometabolome

sampling was done by immediate filtration of the culture broth and the supernatant was stored at $-20\,^{\circ}C$ until analysis.

**Exometabolome analysis.** Extracellular glucose, ethanol, pyruvate, succinate, and acetate were quantified using an HPLC system (ultimate 3000 HPLC, Thermo Fisher, Waltham, MA) with a BioRad HPX-87H column (BioRad, Hercules, CA) and an IR detector, with 5 mM $H_2SO_4$ as the elution buffer at a flow rate of 0.6 ml $min^{-1}$, and an oven temperature of $45\,^{\circ}C$.

**RNA sequencing.** RNA was extracted using Qiagen RNeasy Mini Kit (Qiagen, Hilden, Germany) according to manufacturer's protocol. RNA integrity was examined using a 2100 Bioanalyzer (Agilent Technologies, Santa Clara, CA). RNA concentration was determined using a Qubit RNA HS Assay Kit (Thermo Fisher, Waltham, MA). The Illumina TruSeq Stranded mRNA Library Prep Kit (Illumina, San Diego, CA) was used to prepare mRNA samples for sequencing. Paired-end sequencing (MID Output $2 \times 75$ bp) was performed on an Illumina NextSeq 500 (Illumina, San Diego, CA). Reads were quality controlled, mapped to the *S. cerevisiae* reference genome (Ensembl R64-1-1), and counted using the nf-core RNAseq pipeline (SciLifeLab, Stockholm, Sweden), available at https://nf-co.re/rnaseq.

**Quantitative proteome measurements.** All liquid chromatography-MS (LC-MS) experiments were performed on an Orbitrap Fusion Tribrid mass spectrometer interfaced with an Easy-nLC1200 nanoflow LC system (both Thermo Fisher Scientific, Waltham, MA, USA). Peptide and protein identification and quantification was performed using Proteome Discoverer version 2.2 (Thermo Fisher Scientific) with Mascot 2.5.1 (Matrix Science, London, UK) as a database search engine.

The global relative protein quantification between the samples was performed via the modified filter-aided sample preparation (FASP) method[48], which included the two-stage digestion of each sample with trypsin in 1% sodium deoxycholate (SDC)/50 mM triethylammonium bicarbonate buffer and labeling with the TMT 10plex$^{TM}$ isobaric reagents (Thermo Fischer Scientific) according to the manufacturer's instructions. The pooled reference sample was prepared from the aliquots of the lysates of *S. cerevisiae* CEN.PK 113-7D cells from the J. Nielsen Lab (Chalmers, Gothenburg, Sweden) and processed alongside the eight samples from the nitrogen limitation conditions. The combined TMT-labeled set was pre-fractionated into 20 final fractions on an XBridge BEH C18 column (3.5 μm, 3.0 × 150 mm; Waters Corporation, Milford, MA, USA) at pH 10 and each fraction was analyzed using a 60 min LC-MS method. The most abundant peptide precursors were selected in a data-dependent manner, collision-induced dissociation (CID) MS$^2$ spectra for peptide identification were recorded in the ion trap, the seven most abundant fragment ions were isolated via the synchronous precursor selection (SPS), fragmented using the higher-energy collision dissociation (HCD), and the MS$^3$ spectra for reporter ion quantification were recorded in the Orbitrap.

IBAQ approach[7] was used to estimate the absolute protein concentrations in the pooled reference sample. An aliquot of 50 μg of the pooled sample was spiked with 10.6 μg of the UPS2 Proteomics Dynamic Range Standard (Sigma-Aldrich, Saint-Louis, MO) and digested using the FASP protocol, pre-fractionated at pH 10 on the XBridge BEH C18 column (3.5 μm, 3.0 × 150 mm) into 10 fractions, and each fraction was analyzed 3 times using a 90 min method with MS$^1$ spectra recorded at 120,000 resolution, and the data-dependent CID MS$^2$ spectra recorded in the ion trap with 1 s duty cycle. The label-free data were processed using the Minora feature detection node in Proteome Discoverer version 2.2 and the quantitative values from three technical (injection) replicates were averaged. Forty-three proteins from the UPS2 standard were detected with two or more unique peptides and used to calculate the linear regression coefficients between the known concentrations of the UPS2 proteins and their corresponding iBAQ measurements. The slope and *y*-intercept of the linear regression were used to quantify the yeast proteins in the pooled reference sample. The adjusted $R^2$ of the linear model was 0.95, $p = 2.2e - 16$. The absolute concentration estimates were calculated for each of the eight samples using the iBAQ-based absolute values for the pooled reference sample and the relative abundance values from the TMT experiment.

For the validation of RP abundance, 51 peptides were chosen and synthesized as the SpikeTides TQ peptide standards by JPT Peptide Technologies (Berlin, Germany). The pooled reference sample and the mixture of the equal amounts of the standard peptides were digested with trypsin. Four aliquots of the standard peptide mixture containing 125 fmol, 500 fmol, 2.5 pmol, and 12.5 pmol of each peptide were labeled with the TMT 10plex$^{TM}$ reagents 128C, 129N, 130C, and 131, respectively. The digested reference sample was divided into the ≈4 μg and 25 μg aliquots, and labeled with the reagents 127N and 126. The combined and C18-purified TMT sample was analyzed using the 140 min LC-MS methods that featured the targeted inclusion list with the *m/z* values and retention times of the labeled synthetic peptides, CID MS$^2$ for peptide identification and SPS-HCD-MS$^3$ with the maximum ion injection time of 250 ms and an enhanced AGC target of 1e6 for reporter ion quantification. The quantification spectra were inspected manually, and the S/N reporter abundances from the high-intensity spectra

without visible peptide interference were selected. The four known concentrations of the synthetic standard peptides formed a mini-calibration curve, to ensure that the concentration-to-signal response is linear and that the intensity of the same peptide in the cell lysate sample falls within the linear range in each quantification spectrum. Overall, 49 synthetic peptides passed quality control. The absolute concentration of each peptide in the reference sample was then calculated using the slope of the S/N signal-vs.-concentration linear regression on the synthetic standards and the S/N intensity of the peptide in the reference. When considering RP paralogs separately, common peptides that can be mapped to either paralog were included in the quantification of both paralogs, to avoid under-estimating the total ribosome abundance. When paralogs were summed, common peptides were added only once.

The detailed experimental procedures, LC-MS, and data processing parameters are described in the Supplementary Methods.

**Data processing and analysis.** For transcriptomics, the absolute concentrations of 31 transcripts with >10 FPKM (fragments per kilobase of transcript per million mapped reads), and covering the entire dynamic expression range, were measured using lysates of *S. cerevisiae* CEN.PK 113-7D cells from the J. Nielsen lab (Chalmers, Gothenburg, Sweden). Linear regression between the absolute concentrations of these mRNAs and their corresponding FPKM values from RNAseq were performed to obtain the slope and y-intercept, which were used to quantify all mRNA in this study. The adjusted $R^2$ of the linear model was 0.84, $p = 2.6e - 13$. The calculated mRNA abundance was then scaled to the total RNA content measured by Qubit RNA HS Assay Kit (Thermo Fisher, Waltham, MA). In estimations of ribosome abundance, 80% of the total RNA is assumed to be rRNA[49].

For proteomics, detailed data processing parameters are described in the Supplementary Methods. For transcriptome and proteome allocation to 99 GO-slim terms, genes assigned to each GO-slim term can be found on the Saccharomyces Genome Database[10], available at: https://downloads.yeastgenome.org/curation/literature/go_slim_mapping.tab.

**Flux balance analysis and flux variability analysis.** The consensus yeast metabolic model Yeast8.1 (https://github.com/SysBioChalmers/yeast-GEM) was converted to enzyme-constrained ecYeast8.1 by GECKO[18]. MATLAB R2016b (MathWorks, Inc., Natick, MA) with Gurobi solver (Gurobi Optimizer, Beaverton, OR) in the COBRA toolbox[50] was used for the simulations. Condition-dependent biomass composition was introduced into the model by scaling the coefficients of the protein and RNA pseudo-reactions to equal the measured protein and RNA abundance, and scaling the carbohydrate coefficient to maintain an equivalent amount of mass in the biomass pseudo-reaction[18]. The growth-associated maintenance, which reflects largely the protein and polysaccharide polymerization costs, was re-calculated as previously performed[18]. Each model was then constrained by the metabolic enzyme abundance, measured exchange fluxes, and growth rate (Supplementary Data 6). In FVA, the boundaries of the solution space for each flux was calculated by, in turn, setting the objective function to maximize and minimize each flux. The most-likely value of each flux, calculated as the mode of the distribution of 1000 random samplings of a pair of randomly weighted objective functions, was then taken as the optimal solution[23], as implemented in the RAVEN toolbox[51]. To calculate the minimum enzyme demand, the objective function was set to minimize enzyme pool pseudo-metabolite $P_{met}$. The overall enzyme saturation coefficient ($\sigma$) was calculated as the in-silico simulated minimum enzyme demand divided by the in-vivo measured enzyme abundance. In ecGEM simulations, each enzyme has an exchange "flux," which is the pseudo-reaction reflecting the amount of a given enzyme pulled by the model from the total enzyme pool pseudo-metabolite $P_{met}$, and has a unit of mmol $gDW^{-1}$[18]. Individual enzyme usage was calculated as the in-silico simulated enzyme exchange "flux" divided by the in-vivo measured enzyme abundance. Usage of superpathways[10] was calculated as the sum of enzyme exchange divided by the sum of enzyme abundance for all enzymes in a given superpathway. Superpathways containing at least five genes and with non-zero simulated flux were examined.

**Reporting summary.** Further information on research design is available in the Nature Research Reporting Summary linked to this article.

## Data availability

Processed quantitative transcriptomics and proteomics data are in Supplementary Data 1. Raw RNAseq data are available at ArrayExpress, accession E-MTAB-8245 [https://www.ebi.ac.uk/arrayexpress/experiments/E-MTAB-8245/]. The mass spectrometry proteomics data has been deposited to the ProteomeXchange Consortium via the PRIDE[52] partner repository with the dataset identifiers PXD12803 [http://www.ebi.ac.uk/pride/archive/projects/PXD012803] for the IBAQ dataset, PXD014962 [http://www.ebi.ac.uk/pride/archive/projects/PXD014962] for the TMT-based relative quantification dataset, and PXD015025 [http://www.ebi.ac.uk/pride/archive/projects/PXD015025] for the absolute quantification experiment with the standard peptides. GO-slim term are available from the Saccharomyces Genome Database [https://downloads.yeastgenome.org/curation/literature/go_slim_mapping.tab]. All other supporting data are available from the corresponding author on request.

## Code availability
Custom code is available from the corresponding author on request.

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

## Acknowledgements
We thank Avlant Nilsson (Chalmers University, Sweden) for discussions. We thank Pannipa Pornpitakpong and Alexandra Hoffmeyer (DTU, Denmark) for conducting RNA sequencing. The Proteomics Core Facility at the University of Gothenburg is grateful to the Inga-Britt and Arne Lundbergs Forskningsstiftelse for the donation of the Orbitrap Fusion Tribrid MS instrument. This research was supported by funding from the Novo Nordisk Foundation (grant number NNF10CC1016517) and the Knut and

Alice Wallenberg Foundation. Open access funding is provided by Chalmers University of Technology.

## Author contributions

R.Y. and J.N. conceived the study. R.Y. designed the experiments. R.Y., K.C., R.P., J.B., Q.Q., E.V., and C.S. performed experiments. R.Y. and E.V. analyzed data. All authors wrote the manuscript.

## Competing interests

The authors declare no competing interests.
