## [Peer Review File · Nature Communications]

Reviewers' comments:

Reviewer #1 (Remarks to the Author):

In this study, the authors estimated reserves of cellular capacities by analyzing allocations in transcriptome and proteome of yeast cells cultivated in nutrient-limiting chemostat cultures. They reported many interesting findings: 1) Within the same growth rate regime, allocations in transcriptome and proteome are conserved at the process level. 2) By limiting nitrogen in the growth medium, they successfully created cellular state in which transcriptome and proteome is fully economized. 3) The allocation of the proteome is compensated within each process, while the expression level of each protein is changed upon N-limitation. 4) A large number of metabolic proteins are reserved under C-limitation. 5) Translational efficiencies of metabolic proteins are increased upon N-limitation, while translational efficiencies of proteins in the translation are constantly high. 6) Sub-stoichiometric ribosomal proteins are increased upon N-limitation suggesting that in the economical state, those ribosomal proteins are used for efficient translation of metabolic proteins. All these findings and arguments in this study are quite novel and stimulating to understand the cellular economy. I, however, have some concerns and suggestions that will strengthen and improve the arguments in this study.

Major comments

1) It is better to attach a diagram explaining/summarizing what happens or is expected for cellular physiology in each nutrient-limiting condition. For example, the authors assume that in the C-limiting condition yeast cells are in the respiro-fermentation state, while in N-limiting conditions they are fermentation state. Those cellular states might be assumed by their transcriptome/proteome data. Also, could the author provide a diagram summarizing their findings?

2) As far as I recognize, they showed for the first time that there is a large reserve or overcapacity of metabolic enzymes. I think the authors should emphasize and discuss this part in more detail. Do the authors think that this happens in C-rich conditions? If it is, could the authors propose previous works or further experiments supporting their hypothesis? I am wondering if this reserve is associated with a transient state created by the imbalance between C-limitation and N-limitation, but it might not happen C-rich conditions. What do the authors think about this?

3) The authors claimed that translational efficiencies of metabolic enzymes are kept low in the C-limiting conditions because of some translational regulation. However, I can imagine an alternative possibility– mRNAs are too many compared with the ribosome number. Suppose that, if 100 ribosomes are fully-engaged in the translation of 100 mRNAs in the economical state, increasing mRNA levels to 200 mRNAs under a fixed ribosomal number (100 ribosomes) will reduce superficial translational efficiency. Therefore, changes in translational efficiencies are created not by regulation but just by iteration of ribosomes. Is this an impossible idea?

4) Their allocation analysis shown in Figure 1b used GO categories in which many genes are overlapped. In Table S2, the total number of genes is 13,415, indicating that one gene is categorized into several categories. Doesn't this become bias to create a better correlation in their allocation analysis? Also, this might make the difference in the correlations between Figure 1a and Figure 1b. How many independent genes are analyzed in Figure 1b? How does the correlation become when the analysis in Figure 1a is performed only using the genes categorized in Figure 1b (or Table S2). Moreover, gene names categorized in Table S2 should be presented, so that readers can reproduce the author's results.

5) The authors claimed that the resource allocation varies with cell growth rate by referring to ref#4 (Metzl-Ras 2017). Is it reproduced by their own analysis? They can compare the allocation between the growth rates of 0.2 h⁻¹ (their data) and 0.1 h⁻¹ (ref#7, Lahtvee 2017). Also, they can use gene categories used in ref#4 to confirm the authors' conclusion.

6) They claimed that total proteome allocation remains unchanged within each pathway, despite the abundance of individual enzymes are changed, by adjustments of other enzymes (Figure 2e). However, a big expression change of a component whose expression level is low has a little effect on the sum of all components if expression levels of components are distributed in log-scale. What happens when contribution rates of changed proteins are considered, or simply making a graph without ADH2 of Figure 2e.

7) Could the authors show which enzymes make the large differences in the pathway usages shown in Figure 3b? Enzymes in methionine biosynthesis, sulfur amino acid biosynthesis, and threonine and methionine biosynthesis are overlapped, and changes in the pathway usage seem to be explained by the dramatic expression change in Met17. Also, dramatic changes in the usages of other pathways might be explained with the expression changes in only a small number of enzymes, which could be a hub to control the activity of the whole pathway.

8) The authors claimed that cellular metabolisms in the C-limiting and N-limiting conditions are different; respiratory and respiro-fermentative metabolisms. If so, mitochondrial transcriptome and proteome should be affected. In fact, as far as I see, expression-levels of Cox6 and Cox9 are dramatically reduced upon N-limitation (Table S1). Why these changes are not reflected in the allocation change shown in Figure 1c?.

9) The following argument is my impression of their results, and I just want to know their opinion, and the authors do not have to reflect their opinion on their paper. Their findings that 50% of the proteome, 75% of the transcriptome, 30% of metabolic capacity, and 50% of translational capacities serves as a reserve, are surprising and interesting. However, do they happen because of the imbalance between C-source and N-source, or under C-limiting? N-source might be more important for the cell, and the cell might accumulate N-source under N-rich conditions as a form of protein or nucleic acid. Or, because of C-limiting, yeast cells are waiting for the expected C-source fluctuation (which might often happen in their growth environments), and the observed state might not be the result of regulation but a transient state usually not happens. Namely, this reserve might not serve the cellular robustness that is implemented into the cellular system, but a transient state created by the imbalance of C-source and N-source.

Is it possible that the observation that enzymes in CCMs are excess is because yeast cells are just waiting for coming glucose? In the natural conditions, I do not think that the yeast cells are kept in continuous C-limiting conditions. C-source should be more fluctuating, and N-source is more limiting and they need to accumulate N-source in the cells. The reserves might not be a result of prepared regulation but created by the imbalance between bio-mass production and energy production. Such a metabolic imbalance state might be a similar state in metabolic syndromes in humans. I do not think the authors need to discuss this in their paper, but I would like to ask what the authors think about these ideas.

Minor comments

1) What are the scales of the vertical axis in Figures 1c and 1d?

2) Lines 219-222, I do not understand the authors' logic. What part of previous sentences is consistent with the higher propensity of internal adjustment in metabolic pathways?

3) Line 285-302 contains many discussions and seemed not to be suitable for the Results section.

Reviewer #2 (Remarks to the Author):

General impression

In their manuscript "Nitrogen limitation reveals large reserves in metabolism and translational capacities of yeast", Yu et al. report proteomics and transcriptomics measurements of yeast chemostat cultures under varying nitrogen supply, implying varying extracellular glucose levels and a fixed growth rate. The data indicate massive changes in cells' overall protein and mRNA content, but relatively little re-allocation of protein and mRNA resources between gene functional categories.

The authors observe strong changes in enzyme efficiencies (i.e., a changing ratio between fluxes and catalyzing enzymes) and in translational capacities (i.e. changing gene-specific protein/mRNA ratios). The latter effect is explained by a changing composition of ribosomes. From their data, the authors conclude that yeast cells, under glucose limitation maintain large reserve capacities (both in metabolism and translation).

The paper is clearly structured and well written. The experimental approach is sound and of high quality, and the data will be highly valuable for cell biologists and for modellers. A specifically interesting aspect of this work is that it studies the re-allocation of protein and RNA resources at a constant growth rate, thus highlighting the different effects of growth rate changes vs (growth-rate neutral) environmental changes on cellular resource allocation.

----- Major remarks

(1) Interpretation in terms of "Reserve capacity"

The authors claim that cells maintain a "reserve capacity". For example, they write: "This revealed that 75% of the total transcriptome and 50% of the proteome are produced in 50% excess of what is necessary to maintain growth." and state: "Remarkably, several metabolic pathways accommodated for an increased flux with decreased enzyme abundance, clearly indicating that cells maintain large reserves in their metabolic capacity." Their wording suggests that the cells would in theory be able to sustain growth using much lower transcript and protein levels under the given conditions. For example, the authors state that "enzyme usage was fully economized under these conditions" (implying that enzyme levels could have been lower in other conditions, too) and mention a "minimum enzyme demand" (again, implicitly suggesting that this is actually an enzyme demand under the condition in question, not just a theoretical demand achievable under other, more favorable conditions).

I'm not sure if this claim is directly supported by the data. At least, I see a second interpretation that could explain away a part of the "reserve capacity": glucose limitation may force cells to use higher enzyme levels because of less favourable kinetics and thermodynamics; in this interpretation, nitrogen limitation in a chemostat would be a more favorable condition, because the resulting higher glucose levels lead to higher supply of carbon and energy.

Thus, the enzyme/flux ratio under glucose limitation would not be higher because cells maintain a reserve, but simply because higher enzyme levels are needed to maintain the required fluxes under this condition. Calling the higher enzyme amounts a "reserve" would be like saying that cars in a traffic jam maintain a "speed reserve" (as "proven" by the fact that the same cars have been observed to drive faster at other times).

I have the same criticism about other works (e.g. by the Palsson group) that use FBA (or similar methods) and ignore kinetic and thermodynamic effects (i.e., the dependence of enzyme efficiency on metabolite levels). If enzyme kinetics is ignored, enzymes appear to "remain below their maximal capacity", which the model cannot explain; but instead of calling them "unexplained", Palsson in one of his papers called this protein fraction "unused", and it may be that the authors

now re-interpret this "unused enzyme fraction" as a "reserve capacity".

According to kinetic models, this "unused" protein fraction can be quite high and depend on external conditions. Risking to disclose my identity, I point the authors to Wortel et al. (2018), PLoS Comput Biol 14(2):e1006010, Figure 11 in the supplementary text, where optimal enzyme capacity utilisation was studied under the constraints of a kinetic metabolic model. This optimal, yet not "ideal" enzyme capacity utilisation would already predict a large "reserve capacity" (when interpreted wrongly).

To avoid misunderstandings: I do not doubt that cells keep reserves to anticipate changes. But I'm not sure that this is the right (or only relevant) interpretation of the data presented. Therefore, I suggest to discuss both possible interpretations and to explain what a "reserve" means precisely, or to show that my interpretation can be ruled out. Maybe, the data in Figures 2 d, e, and f, together with Supplementary table S3 ("Measured metabolic flux") will help prove me wrong.

In any case, I think that the paper will profit from discussing (and possibly discarding) my alternative interpretation, especially if the aim is to "streamline biosynthetic processes for synthetic biology applications" (where the actual possibility of reducing cellular burdens will play a role).

(2) Biomass composition

If the protein and RNA amounts per cell dry weight are decreasing (as shown in Figure 2a), it would be interesting to know what happens to the rest of the cell mass - do the ratios of lipids, cell wall, and small metabolites remain the same. Could you briefly discuss this point?

Minor remarks

70: is the Pearson correlation computed for a single condition, across genes, or for lumped data across genes and conditions? Please clarify.

78: "high correlation with the number of genes in each GO-slim term 78 (Pearson $r = 0.73$), while allocation of the proteome did not (Pearson $r = 0.35$, Fig S3a-b)" - the difference between 0.78 and 0.35 is rather gradual, so the distinction "high correlation VS not" can be a bit misleading

85/86: the terms "rich" and "poor" in allocation sound a bit odd to me - I see allocation as a cost, so the authors' wording sounds like calling somebody "rich in debts"

100: "49% carbon and 9% nitrogen" I imagine that these numbers change under varying carbon / nitrogen limitation; it would be important to mention this, especially since the changes in RNA and protein amounts observed in the paper will probably have an effect on the elemental composition of the cell.

151 "However, there is a paradoxical negative correlation between the abundance of enzymes in CCM and μ " I'm not sure this is paradoxical. If enzymes work more efficiently (i.e., higher flux per enzyme molecule due to favorable substrate and product concentrations), the number of enzyme molecules can be reduced, freeing resources and allowing the cell to grow faster. Or am I missing something?

158 "We therefore constrained the ecGEM of yeast, ecYeast8.1, with an upper bound for the exchange reaction of the enzyme pool pseudo-metabolite equal to Pmet." Please rephrase / explain

162 "with the objective function of minimizing the enzyme pool exchange reaction." Please rephrase / explain

163 "that corresponds to minimum enzyme demand" Please clarify: do you mean a minimum enzyme demand under the conditions in question (e.g. glucose limitation) and the resulting intracellular metabolite concentrations, or a minimum enzyme demand under the most ideal, favorable conditions (but possibly not achievable under the conditions in question) ?

167 "confirming that enzyme usage was fully economized under these conditions." maybe rephrase? "confirming that enzymes work at their maximal possible speed under these conditions."

268: does the molecule number refer all 54 core subunits together, or to each of the 54 types?

269: the same, for the non-core RPs

(same for legend Figure 5)

271: "parabolically" is unclear to me; maybe "quadratically"?

292: "Benchmarking the quality of this gene set is the inclusion .." -> syntax not clear to me

315: "maximising cell economics" Please rephrase more specifically

323: "while growth environment determines the gene expression ratio within a given allocation scheme." -> "Allocation scheme" is not clear to me

339: analyses -> analysis

468: "The optimal solution was then taken as the most-likely value of each flux, calculated as the mode of the distribution of 1,000 random samplings of a pair of randomly-weighted objective functions" -> not fully clear to me; do you mean "The most-likely value of each flux, calculated as the mode of the distribution of 1,000 random samplings of a pair of randomly-weighted objective functions, was then taken as the optimal solution"?

629 "Relative # of GO-slim terms belonging in each functional category with FDR-adjusted p Fisher < 0.05 are shown" just to be sure - you mean "shown on the y axis"?

663: "Minimum demand for the enzyme pool was calculated with the enzyme saturation coefficient σ simulated by ecYeast8.1." please explain

667: "(mode of distribution of the protein exchange reactions in random sampling)" please explain

694: "RP:rRNA ratio for each RP" please clarify: is this the RP amount (for a specific RP) divided by the total rRNA amount in the cell?

Fig. 4 Please avoid the terms "early", "intermediate", and "late" modulation, suggesting a temporal order (whereas the ordering starting from C-lim and ending with C/N=115 is arbitrarily chosen)

Reviewer #3 (Remarks to the Author):

In this study, the authors systematically reduce nitrogen (using chemostat cultures of yeast), to discover that cells have enormous reserves in translation (and likely metabolic) capacities.

Overall, this is an excellent systems-level study, well executed experimentally and computationally, and with many unexpected findings. The key take-home is that cells have very

large buffers of (especially) translation capacity, and some metabolic capacity, as nitrogen availability decreases. I would consider this a stand-out study in this space, since it addresses many limitations of other important studies that address systems-level responses to different nutrient limitations. Specific prior studies I would mention are Brauer MJ et al 2008, which only considered gene expression, growth rate and metabolism in different nutrient limitations, to come up with predictable growth models, (2) Boer VM et al 2010, which defines what metabolites are most limiting for a cell (and which also identified some metabolic buffers or reserves), (3) Kafri et al 2016 and MetzI-Raz et al 2017, which definitively identify excess translation capacity in rapidly growing cells. This current study is a substantial advance over all these studies, while nicely complementing many aspects of these. (Also, I would consider citing the first two studies, which are very relevant to this study).

After some minor clarifications, as well as a few points that can either be included in the results or raised in the discussion, this manuscript should be published without much delay.

i) It will help if in the results section itself, a little more detail on each steady state growth condition is provided (so that C/N=30, C/N=50 etc can be made clear upfront). Here, adding a line or two to clarify at what dilution rate did the cell doubling-time decreased (and by how much) will make it far easier to read.

ii) The (poor/average) correlation of the entire transcriptome with the proteome is consistent with many prior studies. A few of them should be cited here.

iii) A slightly more detailed break-down of the relationship between transcriptome & proteome allocation to each process would be useful. Here, the overall correlation to 99 GO-slim terms (across 6 broad groups) are shown. If each group (e.g. metabolism, with 18 groups in it) is taken, broken into a larger set of GO terms, and correlates drawn between transcripts and protein, what would the r^2 look like? This is mostly to ask if in this space, there are some groups that correlate very well between transcript and protein (I would guess cell cycle and organelle, or DNA maintenance for example), while others might show some surprises.....and may be expand on this idea of translation and metabolic reserves (some groups in metabolism where for example the proteome might be buffered or unchanged with nutrients). Also, with respect to metabolism, it might reveal that the proteomes of some key metabolic node enzymes are substantially de-linked from transcriptional control (which is likely, based on what we know about metabolism). Its worth doing this exercise (and including these as supplements) if something striking shows up. This will help substantiate the authors' statement ".....with processes "rich" in transcriptomic allocation having an even greater proteomic allocation, while processes "poor" in transcriptome allocation would be repressed at the protein level...."

iv) A very interesting comparison, now possible with this study, is to separate metabolic enzymes into what are known to be/classified as rate-limiting (or sometimes pathway entry) steps, vs. those within a pathway. This can especially be done for the larger metabolic pathways (eg. some pathways in amino acid biosynthesis). Here, do some of these rate-limiting step enzymes show a greater change in both transcript and protein, with nitrogen-content reduction? Does this also correlate with glucose repression (like the Adh2 example observed)?

v) Related to figure 3: estimating total enzyme reserves, and decoupling/remove changing growth rates (by using a minimum enzyme demand criteria) is nice (and appropriate). But in part the reason there is a negative correlation between growth rate and the enzyme amounts is that the growth rate directly depends on the metabolic flux, but (as described in Hackett et al 2016), this flux depends most strongly on substrate concentrations (which change when for example nitrogen is reduced), and not just on enzyme amounts. This also allows you to justify your FBA approach (which will only consider steady-state flux, but doesn't fully account for some metabolite that is rapidly changing). So more clearly acknowledging/restructuring your conclusions, and explaining your analysis are important.

vi) The observation that translation reserves are used to preferentially translate metabolic proteins is striking (related to Fig 4C). First, how much of this correlates purely to transcript abundance (and how many of the metabolic transcripts preferentially translated had the highest transcript amounts)? If this is a linear correlation, this should be noted and explained. Also, are there any notable outliers (relatively low transcript, but reserve translation was used to translate these? Also, currently, the authors do not suggest any clear mechanism for how this might happen.

vii) Relatedly (and also related to Figure 2), what can we learn about enzyme paralogs from these data? Yeast famously have several paralogs (due to the genome duplication), and many of these are considered "minor isoforms" etc. With changes in nitrogen, do the authors note a strong switch between the expression of paralogs of metabolic genes (e.g. GDH1 and GDH2, both very relevant as nitrogen decreases, in the synthesis of glutamate from alpha-KG)? This is something the authors have not probed, but might be the perfect dataset to get insight into the roles/regulations of paralogs with changes in nutrient availability. This also can extend to the use of isozymes (and not paralogs) in metabolic reactions. If the authors note interesting correlations, their data might explain a very long standing question on the role of these paralogs, and why they are retained. Notably, the older data from the yeast metabolic cycles (Tu et al 2005 science) clearly show distinct paralog expression at different metabolic states, which is relevant.

viii) Along with clear changes in ribosomal subunit stoichiometry, there does seem to be some changes in proteasomal subunit stoichiometries (with decreasing nitrogen). Also (see point above), with the ribosomal genes, a clear correlation with paralog specific responses are seen. This deserves some better explanation. Even some speculation in the discussion on how this might be so (and how to think of this) would be very useful.

A point you raise in the discussion is that different RP compositions might explain the selective translation of metabolic genes. This is a nice hypothesis, but with the current data is too much of a stretch.

A few other points that can be included in the discussion:

- its clear that while in the laboratory we use very large excesses of carbon and nitrogen, cells are far more robust and used to a lot 'less'. So what is 'normal' carbon & nitrogen in medium for cells? What is 'normal' medium?

- to me its clear that the metabolic enzyme buffer is huge, so overrelying on changes in transcript of metabolic enzymes (to make major interpretations) is troublesome. This warrants a discussion point.

very minor comments:

1) Quite impressed with how well the proteomics has been done quantitatively here. The reproducibility of the biological replicates is impressive.

2) Was able to access and look at the transcriptomics data in the repository.

But the same was not possible for the proteomics data, with the link provided being inactive ('pride' repository <https://www.ebi.ac.uk/pride/archive/login>). However, since the experimental details provided for all the proteomics are excellent, I won't need to look at the raw data. I was curious about some of the relative peptide amounts for some of the ribosomal subunits, and wanted to look at those, as well as the IBAQ quantifications. It is nice that the authors have fully clarified their 'manual inspection' and selection method for spectra, and their approach is quite reasonable here.

Reviewer #1 (Remarks to the Author):

In this study, the authors estimated reserves of cellular capacities by analyzing allocations in transcriptome and proteome of yeast cells cultivated in nutrient-limiting chemostat cultures. They reported many interesting findings: 1) Within the same growth rate regime, allocations in transcriptome and proteome are conserved at the process level. 2) By limiting nitrogen in the growth medium, they successfully created cellular state in which transcriptome and proteome is fully economized. 3) The allocation of the proteome is compensated within each process, while the expression level of each protein is changed upon N-limitation. 4) A large number of metabolic proteins are reserved under C-limitation. 5) Translational efficiencies of metabolic proteins are increased upon N-limitation, while translational efficiencies of proteins in the translation are constantly high. 6) Sub-stoichiometric ribosomal proteins are increased upon N-limitation suggesting that in the economical state, those ribosomal proteins are used for efficient translation of metabolic proteins. All these findings and arguments in this study are quite novel and stimulating to understand the cellular economy. I, however, have some concerns and suggestions that will strengthen and improve the arguments in this study.

Major comments

1) It is better to attach a diagram explaining/summarizing what happens or is expected for cellular physiology in each nutrient-limiting condition. For example, the authors assume that in the C-limiting condition yeast cells are in the respiro-fermentation state, while in N-limiting conditions they are fermentation state. Those cellular states might be assumed by their transcriptome/proteome data. Also, could the author provide a diagram summarizing their findings?

Under C-limitation yeast cells are respiratory; under N-limitation they are respiro-fermentative. This comes from the r_{CO_2} and $r_{Ethanol}$ measurements, which are provided in Table S4. We have re-written a few sentences in this section (lines 216-218) to help clarify this. We have provided a graphical abstract of our major findings in Fig S23.

2) As far as I recognize, they showed for the first time that there is a large reserve or overcapacity of metabolic enzymes. I think the authors should emphasize and discuss this part in more detail. Do the authors think that this happens in C-rich conditions? If it is, could the authors propose previous works or further experiments supporting their hypothesis? I am wondering if this reserve is associated with a transient state created by the imbalance between C-limitation and N-limitation, but it might not happen C-rich conditions. What do the authors think about this?

The reviewer's concern, here and in point #9, is that perhaps our observed metabolic reserve is a response to low carbon and not a general feature of the cell economy. This is an interesting point because, when C availability is increased (which, when C is the limiting nutrient, allows for faster growth), the cells allocate less of their proteome to CCM – which would seem to indicate that when C is more available, this metabolic reserve is reduced. This has been shown in *E. coli* (Peebo et al 2015, Valgepea et al 2013), and our lab has unpublished data in *S. cerevisiae* showing the same phenomenon. However, when N is the limiting nutrient and we step-wise increase N availability, we also observe a reduction in proteomic allocation to CCM, even though glucose is in excess in these conditions (our unpublished data). At the transcriptomic level, this has also been shown for a number of other nutrient limiting conditions, including in Leu/Ura-limited auxotrophic cells (Brauer et al 2008). So cells maintain this metabolic reserve even in conditions where C (or both C and N) is

in excess, indicating that this is indeed a general feature of the cell economy. We have added this discussion on lines 386-394.

3) The authors claimed that translational efficiencies of metabolic enzymes are kept low in the C-limiting conditions because of some translational regulation. However, I can imagine an alternative possibility—mRNAs are too many compared with the ribosome number. Suppose that, if 100 ribosomes are fully-engaged in the translation of 100 mRNAs in the economical state, increasing mRNA levels to 200 mRNAs under a fixed ribosomal number (100 ribosomes) will reduce superficial translational efficiency. Therefore, changes in translational efficiencies are created not by regulation but just by iteration of ribosomes. Is this an impossible idea?

Yes this is possible – we have added a line to address this (lines 433-434).

4) Their allocation analysis shown in Figure 1b used GO categories in which many genes are overlapped. In Table S2, the total number of genes is 13,415, indicating that one gene is categorized into several categories. Doesn't this become bias to create a better correlation in their allocation analysis? Also, this might make the difference in the correlations between Figure 1a and Figure 1b. How many independent genes are analyzed in Figure 1b? How does the correlation become when the analysis in Figure 1a is performed only using the genes categorized in Figure 1b (or Table S2). Moreover, gene names categorized in Table S2 should be presented, so that readers can reproduce the author's results.

To address this issue of genes being assigned to multiple GO-slim terms, and as suggested by point #5, we used the gene categories in MetzI-Raz et al 2017 where each gene is only assigned to one category (Fig S3, lines 81-83). The correlation was 0.96-0.99, confirming our conclusion. We have indicated in the methods section where the gene names in Table S2 can be found (line 561): https://downloads.yeastgenome.org/curation/literature/go_slim_mapping.tab.

5) The authors claimed that the resource allocation varies with cell growth rate by referring to ref#4 (MetzI-Ras 2017). Is it reproduced by their own analysis? They can compare the allocation between the growth rates of 0.2 h⁻¹ (their data) and 0.1 h⁻¹ (ref#7, Lahtvee 2017). Also, they can use gene categories used in ref#4 to confirm the authors' conclusion.

We have unpublished data confirming that resource allocation varies with cell growth rate in *S. cerevisiae*. This has also been shown in *E. coli* – see Peebo et al 2015 and Valgepea et al 2013, which we cite. The difference between 0.2 h⁻¹ and 0.1 h⁻¹ is not large enough to make the comparison to Lahtvee 2017 meaningful. We have added the analysis using gene categories in MetzI-Ras 2017 (Fig S7, lines 134-136) and it indeed confirms our conclusion.

6) They claimed that total proteome allocation remains unchanged within each pathway, despite the abundance of individual enzymes are changed, by adjustments of other enzymes (Figure 2e). However, a big expression change of a component whose expression level is low has a little effect on the sum of all components if expression levels of components are distributed in log-scale. What happens when contribution rates of changed proteins are considered, or simply making a graph without ADH2 of Figure 2e.

Graphing Fig 2e without ADH2 shows that total allocation is indeed hardly affected. We have therefore removed this point (lines 146-147).

7) Could the authors show which enzymes make the large differences in the pathway usages shown in Figure 3b? Enzymes in methionine biosynthesis, sulfur amino acid biosynthesis, and threonine and methionine biosynthesis are overlapped, and changes in the pathway usage seem to be explained by the dramatic expression change in Met17. Also, dramatic changes in the usages of other pathways might be explained with the expression changes in only a small number of enzymes, which could be a hub to control the activity of the whole pathway.

We have provided this data in Table S8 and Fig S11. As can be seen in Table S8, in many cases the reserves are in the form of unused enzymes (e.g. isozymes), for which we provide an additional analysis (Fig S13, lines 232-240). MET17, however, alone accounts for nearly 50% of the reserves in the indicated pathways (Fig S11) (lines 224-231).

8) The authors claimed that cellular metabolisms in the C-limiting and N-limiting conditions are different; respiratory and respiro-fermentative metabolisms. If so, mitochondrial transcriptome and proteome should be affected. In fact, as far as I see, expression-levels of Cox6 and Cox9 are dramatically reduced upon N-limitation (Table S1). Why these changes are not reflected in the allocation change shown in Figure 1c?

As the reviewer pointed out in point 6, since expression levels span several orders of magnitude, the differential expression of one or two small components has little effect on total allocation.

9) The following argument is my impression of their results, and I just want to know their opinion, and the authors do not have to reflect their opinion on their paper. Their findings that 50% of the proteome, 75% of the transcriptome, 30% of metabolic capacity, and 50% of translational capacities serves as a reserve, are surprising and interesting. However, do they happen because of the imbalance between C-source and N-source, or under C-limiting? N-source might be more important for the cell, and the cell might accumulate N-source under N-rich conditions as a form of protein or nucleic acid. Or, because of C-limiting, yeast cells are waiting for the expected C-source fluctuation (which might often happen in their growth environments), and the observed state might not be the result of regulation but a transient state usually not happens. Namely, this reserve might not serve the cellular robustness that is implemented into the cellular system, but a transient state created by the imbalance of C-source and N-source.

Is it possible that the observation that enzymes in CCMs are excess is because yeast cells are just waiting for coming glucose? In the natural conditions, I do not think that the yeast cells are kept in continuous C-limiting conditions. C-source should be more fluctuating, and N-source is more limiting and they need to accumulate N-source in the cells. The reserves might not be a result of prepared regulation but created by the imbalance between bio-mass production and energy production. Such a metabolic imbalance state might be a similar state in metabolic syndromes in humans. I do not think the authors need to discuss this in their paper, but I would like to ask what the authors think about these ideas.

We address this point as a part of our response to point #2. We have also added a small discussion on the natural habitat of yeast on lines 395-400. As for metabolic syndrome, this is a pathological response to excess carbon/energy (due to high lipid levels in the diet and blood) rather than C-limitation, which interestingly could be why high protein diets are commonly suggested to alleviate metabolic syndrome (e.g. Root and Dawson 2013, Sousa et al 2018) – although of course we cannot directly apply our results to studies of metabolic syndrome.

Minor comments

1) What are the scales of the vertical axis in Figures 1c and 1d?

It is # of enriched GO-slim terms – we have added this (lines 760-761).

2) Lines 219-222, I do not understand the authors' logic. What part of previous sentences is consistent with the higher propensity of internal adjustment in metabolic pathways?

We have clarified this sentence (now lines 267-270).

3) Line 285-302 contains many discussions and seemed not to be suitable for the Results section.

We have moved much of this paragraph to the discussion (lines 423-430).

Reviewer #2 (Remarks to the Author):

General impression

In their manuscript "Nitrogen limitation reveals large reserves in metabolism and translational capacities of yeast", Yu et al. report proteomics and transcriptomics measurements of yeast chemostat cultures under varying nitrogen supply, implying varying extracellular glucose levels and a fixed growth rate. The data indicate massive changes in cells' overall protein and mRNA content, but relatively little re-allocation of protein and mRNA resources between gene functional categories. The authors observe strong changes in enzyme efficiencies (i.e., a changing ratio between fluxes and catalyzing enzymes) and in translational capacities (i.e. changing gene-specific protein/mRNA ratios). The latter effect is explained by a changing composition of ribosomes. From their data, the authors conclude that yeast cells, under glucose limitation maintain large reserve capacities (both in metabolism and translation).

The paper is clearly structured and well written. The experimental approach is sound and of high quality, and the data will be highly valuable for cell biologists and for modellers. A specifically interesting aspect of this work is that it studies the re-allocation of protein and RNA resources at a constant growth rate, thus highlighting the different effects of growth rate changes vs (growth-rate neutral) environmental changes on cellular resource allocation.

Major remarks

(1) Interpretation in terms of "Reserve capacity"

The authors claim that cell maintain a "reserve capacity". For example, they write: "This revealed

that 75% of the total transcriptome and 50% of the proteome are produced in 50 excess of what is necessary to maintain growth." and state: "Remarkably, several metabolic pathways accommodated for an increased flux with decreased enzyme abundance, clearly indicating that cells maintain large reserves in their metabolic capacity." Their wording suggests that the cells would in theory be able to sustain growth using much lower transcript and protein levels under the given conditions. For example, the authors state that "enzyme usage was fully economized under these conditions" (implying that enzyme levels could have been lower in other conditions, too) and mention a "minimum enzyme demand" (again, implicitly suggesting that this is actually an enzyme demand under the condition in question, not just a theoretical demand achievable under other, more favorable conditions).

I'm not sure if this claim is directly supported the data. At least, I see a second interpretation that could explain away a part of the "reserve capacity": glucose limitation may force cells to use higher enzyme level because of less favourable kinetics and thermodynamics; in this interpretation, nitrogen limitation in a chemostat would be a more favorable condition, because the resulting higher glucose levels lead to higher supply of carbon and energy.

Thus, the enzyme/flux ratio under glucose limitation would not be higher because cells maintain a reserve, but simply because higher enzyme levels are needed to maintain the required fluxes under this condition. Calling the higher enzyme amounts a "reserve" would be like saying that cars in a traffic jam maintain a "speed reserve" (as "proven" by the fact that the same cars have been observed to drive faster at other times).

I have the same criticism about other works (e.g. by the Palsson group) that use FBA (or similar methods) and ignore kinetic and thermodynamic effects (i.e., the dependence of enzyme efficiency on metabolite levels). If enzyme kinetics is ignored, enzymes appear to "remain below their maximal capacity", which the model cannot explain; but instead of calling them "unexplained", Palsson in one of his papers called this protein fraction "unused", and it may be that the authors now re-interpret this "unused enzyme fraction" as a "reserve capacity".

Accordin to kinetic models, this "unused" protein fraction can be quite high and depend on external conditions. Risking to disclose my identity, I point the authors to Wortel et al. (2018), PLoS Comput Biol 14(2):e1006010, Figure 11 in the supplementary text, where optimal enzyme capacity utilisation was studied under the constraints of a kinetic metabolic model. This optimal, yet not "ideal" enzyme capacity utilisation would already predict a large "reserve capacity" (when interpreted wrongly).

To avoid misunderstandings: I do not doubt that cells keep reserves to anticipate changes. But I'm not sure that this is the right (or only relevant) interpretation of the data presented. Therefore, I suggest to discuss both possible interpretations and to explain what a "reserve" means precisely, or to show that my interpretation can be ruled out. Maybe, the data in Figures 2 d, e, and f, together with Supplementary table S3 ("Measured metabolic flux") will help prove me wrong.

In any case, I think that the paper will profit from discussing (and possibly discarding) my alternative interpretation, especially if the aim is to "streamline biosynthetic processes for synthetic biology applications" (where the actual possibility of reducing cellular burdens will play a role).

Indeed with our data we cannot differentiate whether our observed reserve capacity exists in the form of excess enzymes, or in the form of reduced catalytic efficiency due to kinetic/thermodynamic effects. We have added a discussion on this in the manuscript (lines 376-383).

We note, however, that in two earlier studies (Jansen et al 2005, Machejo et al 2005), when *S. cerevisiae* are evolved for 90-200 generations in carbon-limited chemostats at constant growth rates, the evolved strains showed lowered specific activities (per unit total protein in cell extracts) for most glycolytic enzymes by 2- to 8-fold, compared to the parental strains. This was interpreted as a decrease in the abundance of glycolytic enzymes in the evolved strains – i.e. that in the parental strains, enzyme abundances are kept in excess of demand – which would provide a competitive advantage by reducing the proteomic burden.

If we suppose the scenario that the reviewer has raised, i.e. that high levels of CCM enzymes are not kept as a reserve but are necessary due to unfavourable thermodynamic and kinetic conditions under glucose limitation, then the laboratory evolution results become difficult to interpret. It would appear, in this scenario, that evolution has selected for enzymes that have become less efficient, and the advantage of this would be unclear.

We therefore retain our original interpretation and discuss the reviewer's points as an "alternative" interpretation in the revised manuscript.

(2) Biomass composition

If the protein and RNA amounts per cell dry weight are decreasing (as shown in Figure 2a), it would be interesting to know what happens to the rest of the cell mass - do the ratios of lipids, cell wall, and small metabolites remain the same. Could you briefly discuss this point?

Earlier work (Lillie and Pringle 1980) showed that the glycogen content increases from 2.5% to 15-20% of the cell dry weight, and trehalose increases from 0.3% to 10-15%. Increased lipid content is also suggested by our data, with RQ >1 (Table S4) suggesting lipogenesis. We have added a discussion on this issue in lines 404-411.

Minor remarks

70: is the Pearson correlation computed for a single condition, across genes, or for lumped data across genes and conditions? Please clarify.

For each condition. Fixed (72-74)

78: "high correlation with the number of genes in each GO-slim term 78 (Pearson $r = 0.73$), while allocation of the proteome did not (Pearson $r = 0.35$, Fig S3a-b)" - the difference between 0.78 and 0.35 is rather gradual, so the distinction "high correlation VS not" can be a bit misleading

We have changed the wording (86)

85/86: the terms "rich" and "poor" in allocation sound a bit odd to me - I see allocation as a cost, so the authors' wording sounds like calling somebody "rich in debts"

We have changed the wording (93/94)

100: "49% carbon and 9% nitrogen" I imagine that these numbers change under varying carbon / nitrogen limitation; it would be important to mention this, especially since the changes in RNA and protein amounts observed in the paper will probably have an effect on the elemental composition of the cell.

Fixed (119)

151 "However, there is a paradoxical negative correlation between the abundance of enzymes in CCM and μ " I'm not sure this is paradoxical. If enzymes work more efficiently (i.e., higher flux per enzyme molecule due to favorable substrate and product concentrations), the number of enzyme molecules can be reduced, freeing resources and allowing the cell to grow faster. Or am I missing something?

Fixed, removed paradoxical (170-171)

158 "We therefore constrained the ecGEM of yeast, ecYeast8.1, with an upper bound for the exchange reaction of the enzyme pool pseudo-metabolite equal to P_{met} ." Please rephrase / explain

Fixed 182-187

162 "with the objective function of minimizing the enzyme pool exchange reaction." Please rephrase / explain

Fixed 192-193

163 "that corresponds to minimum enzyme demand" Please clarify: do you mean a minimum enzyme demand under the conditions in question (e.g. glucose limitation) and the resulting intracellular metabolite concentrations, or a minimum enzyme demand under the most ideal, favorable conditions (but possibly not achievable under the conditions in question) ?

It is calculated with the constraints representing the conditions in question (e.g. glucose limitation and dilution rate of 0.2 h⁻¹) although FBA and FVA do not take metabolite concentrations into consideration. We add this on 175-181; this section is re-phrased 191-195

167 "confirming that enzyme usage was fully economized under these conditions." maybe rephrase? "confirming that enzymes work at their maximal possible speed under these conditions."

Fixed 198-200

268: does the molecule number refer all 54 core subunits together, or to each of the 54 types?

269: the same, for the non-core RPs

(same for legend Figure 5)

rephrased 317-319

271: "parabolically" is unclear to me; maybe "quadratically"?

Yes, fixed 321

292: "Benchmarking the quality of this gene set is the inclusion .." -> syntax not clear to me

We have re-worded this 423-430

315: "maximising cell economics" Please rephrase more specifically

Fixed 360

323: "while growth environment determines the gene expression ratio within a given allocation scheme." -> "Allocation scheme" is not clear to me

Fixed 367-368

339: analyses -> analysis

Fixed 416

468: "The optimal solution was then taken as the most-likely value of each flux, calculated as the mode of the distribution of 1,000 random samplings of a pair of randomly-weighted objective functions" -> not fully clear to me; do you mean "The most-likely value of each flux, calculated as the mode of the distribution of 1,000 random samplings of a pair of randomly-weighted objective functions, was then taken as the optimal solution"?

yes – fixed 574-576

629 "Relative # of GO-slim terms belonging in each functional category with FDR-adjusted p Fisher < 0.05 are shown" just to be sure - you mean "shown on the y axis"?

yes, 760-761

663: "Minimum demand for the enzyme pool was calculated with the enzyme saturation coefficient σ simulated by ecYeast8.1." please explain

fixed – details in methods section 576-579

667: "(mode of distribution of the protein exchange reactions in random sampling)" please explain

fixed – details in methods section 572-576

694: "RP:rRNA ratio for each RP" please clarify: is this the RP amount (for a specific RP) divided by the total rRNA amount in the cell?

Yes, fixed 817

Fig. 4 Please avoid the terms "early", "intermediate", and "late" modulation, suggesting a temporal order (whereas the ordering starting from C-lim and ending with C/N=115 is arbitrarily chosen)

fixed

Reviewer #3 (Remarks to the Author):

In this study, the authors systematically reduce nitrogen (using chemostat cultures of yeast), to discover that cells have enormous reserves in translation (and likely metabolic) capacities.

Overall, this is an excellent systems-level study, well executed experimentally and computationally, and with many unexpected findings. The key take-home is that cells have very large buffers of (especially) translation capacity, and some metabolic capacity, as nitrogen availability decreases. I would consider this a stand-out study in this space, since it addresses many limitations of other important studies that address systems-level responses to different nutrient limitations. Specific prior studies I would mention are Brauer MJ et al 2008, which only considered gene expression, growth rate and metabolism in different nutrient limitations, to come up with predictable growth models, (2) Boer VM et al 2010, which defines what metabolites are most limiting for a cell (and which also identified some metabolic buffers or reserves), (3) Kafri et al 2016 and Metzl-Raz et al 2017, which definitively identify excess translation capacity in rapidly growing cells. This current study is a substantial advance over all these studies, while nicely complementing many aspects of these. (Also, I would consider citing the first two studies, which are very relevant to this study).

We have added these references (line 357)

After some minor clarifications, as well as a few points that can either be included in the results or raised in the discussion, this manuscript should be published without much delay.

i) It will help if in the results section itself, a little more detail on each steady state growth condition is provided (so that C/N=30, C/N=50 etc can be made clear upfront). Here, adding a line or two to clarify at what dilution rate did the cell doubling-time decreased (and by how much) will make it far easier to read.

We have added a line to describe each steady state growth condition (line 59-63). All cultures were maintained at the same dilution rate of 0.2 h⁻¹, or a doubling time of 3.5 h.

ii) The (poor/average) correlation of the entire transcriptome with the proteome is consistent with many prior studies. A few of them should be cited here.

Fixed 74-75

iii) A slightly more detailed break-down of the relationship between transcriptome & proteome allocation to each process would be useful. Here, the overall correlation to 99 GO-slim terms (across 6 broad groups) are shown. If each group (e.g. metabolism, with 18 groups in it) is taken, broken into a larger set of GO terms, and correlates drawn between transcripts and protein, what would the r² look like? This is mostly to ask if in this space, there are some groups that correlate very well between transcript and protein (I would guess cell cycle and organelle, or DNA maintenance for example), while others might show some surprises.....and may be expand on this idea of translation and metabolic reserves (some groups in metabolism where for example the proteome might be buffered or unchanged with nutrients). Also, with respect to metabolism, it might reveal that the proteomes of some key metabolic node enzymes are substantially de-linked from transcriptional control (which is likely, based on what we know about metabolism). Its worth doing this exercise (and including these as supplements) if something striking shows up. This will help substantiate the authors' statement ".....with processes "rich" in transcriptomic allocation having an even greater

proteomic allocation, while processes “poor” in transcriptome allocation would be repressed at the protein level....”

We have added this analysis (Table S3, lines 106-116) and found that 4 out of the 6 groups have high correlation between transcriptome and proteome allocation ($r > 0.85$). The two groups with low correlation are cell cycle and stress response, as expected since both of groups of genes are known for PTMs and protein degradation to play a large role in controlling protein expression. The slopes for each group were also calculated (Table S3) and showed clearly that metabolism, translation, and transcription (“rich” in allocation) have larger slopes than cell cycle, stress, and DNA (“poor” in allocation), substantiating our statement as quoted.

iv) A very interesting comparison, now possible with this study, is to separate metabolic enzymes into what are known to be/classified as rate-limiting (or sometimes pathway entry) steps, vs. those within a pathway. This can especially be done for the larger metabolic pathways (eg. some pathways in amino acid biosynthesis). Here, do some of these rate-limiting step enzymes show a greater change in both transcript and protein, with nitrogen-content reduction? Does this also correlate with glucose repression (like the Adh2 example observed)?

We have added this analysis (Fig S12, lines 229-231). Rate-limiting enzymes (RLEs) were found to have significantly ($p < 0.05$) larger difference in proteome allocation between C-limited and N-limited conditions, although the effect size is very small.

v) Related to figure 3: estimating total enzyme reserves, and decoupling/remove changing growth rates (by using a minimum enzyme demand criteria) is nice (and appropriate). But in part the reason there is a negative correlation between growth rate and the enzyme amounts is that the growth rate directly depends on the metabolic flux, but (as described in Hackett et al 2016), this flux depends most strongly on substrate concentrations (which change when for example nitrogen is reduced), and not just on enzyme amounts. This also allows you to justify your FBA approach (which will only consider steady-state flux, but doesn't fully account for some metabolite that is rapidly changing). So more clearly acknowledging/restructuring your conclusions, and explaining your analysis are important.

We have added this acknowledgment on lines 175-181, and provided more details for this analysis in the methods section (566-584).

vi) The observation that translation reserves are used to preferentially translation metabolic proteins is striking (related to Fig 4C). First, how much of this correlates purely to transcript abundance (and how many of the metabolic transcripts preferentially translated had the highest transcript amounts)? If this is a linear correlation, this should be noted and explained. Also, are there any notable outliers (relatively low transcript, but reserve translation was used to translate these? Also, currently, the authors do not suggest any clear mechanism for how this might happen.

We have added this analysis in Fig S16 (lines 280-281). The use of translational reserves is neither correlated with mRNA abundance, nor with changes in mRNA abundance between conditions. We have added a line of speculation as to how this could occur (lines 433-434).

vii) Relatedly (and also related to Figure 2), what can we learn about enzyme paralogs from these data? Yeast famously have several paralogs (due to the genome duplication), and many of these are considered “minor isoforms” etc. With changes in nitrogen, do the authors note a strong switch between the expression of paralogs of metabolic genes (e.g. GDH1 and GDH2, both very relevant as

nitrogen decreases, in the synthesis of glutamate from alpha-KG)? This is something the authors have not probed, but might be the perfect dataset to get insight into the roles/regulations of paralogs with changes in nutrient availability. This also can extend to the use of isozymes (and not paralogs) in metabolic reactions. If the authors note interesting correlations, their data might explain a very long standing question on the role of these paralogs, and why they are retained. Notably, the older data from the yeast metabolic cycles (Tu et al 2005 science) clearly show distinct paralog expression at different metabolic states, which is relevant.

We have added this analysis (Fig S13, Table S9) which showed that out of 173 reactions that can be catalyzed by isozymes, 54 reactions (31%) had at least 1 isozyme DE by >2-fold depending on the culture condition (Fig S13). We note that few isozymes showed “switching” behaviour, e.g. while GDH3 is differentially expressed by several fold (representing 0.1% to 0.4% of total glutamate dehydrogenase expression), GDH1 remained the major isoform (>99%) of this enzyme in all conditions. Similar observations were made for pyruvate kinase: PYK2 (minor isoform, 0.1-0.7%) and CDC19 (major isoform, >99%); and many others (Table S9). See lines 232-240.

viii) Along with clear changes in ribosomal subunit stoichiometry, there does seem to be some changes in proteasomal subunit stoichiometries (with decreasing nitrogen). Also (see point above), with the ribosomal genes, a clear correlation with paralog specific responses are seen. This deserves some better explanation. Even some speculation in the discussion on how this might be so (and how to think of this) would be very useful.

We have added this analysis (Fig S22, lines 342-346). Interestingly we find only changes in subunit stoichiometry in the gate and substrate recognition domains; not in the catalytic subunits or any of the ATPases (Fig S22). Together suggesting that cells become more selective in both protein translation and degradation when nitrogen is limiting.

A point you raise in the discussion is that different RP compositions might explain the selective translation of metabolic genes. This is a nice hypothesis, but with the current data is too much of a stretch.

We have removed this point.

A few other points that can be included in the discussion:

- its clear that while in the laboratory we use very large excesses of carbon and nitrogen, cells are far more robust and used to a lot ‘less’. So what is ‘normal’ carbon & nitrogen in medium for cells? What is ‘normal’ medium?

We have added a discussion on the natural habitat of yeast on lines 395-400.

- to me its clear that the metabolic enzyme buffer is huge, so overrelying on changes in transcript of metabolic enzymes (to make major interpretations) is troublesome. This warrants a discussion point.

We have added a discussion on this on lines 413-415.

very minor comments:

1) Quite impressed with how well the proteomics has been done quantitatively here. The reproducibility of the biological replicates is impressive.

2) Was able to access and look at the transcriptomics data in the repository.

But the same was not possible for the proteomics data, with the link provided being inactive (‘pride’

repository <https://www.ebi.ac.uk/pride/archive/login>). However, since the experimental details provided for all the proteomics are excellent, I won't need to look at the raw data. I was curious about some of the relative peptide amounts for some of the ribosomal subunits, and wanted to look at those, as well as the IBAQ quantifications. It is nice that the authors have fully clarified their 'manual inspection' and selection method for spectra, and their approach is quite reasonable here.

We apologize for this – the link should be <https://www.ebi.ac.uk/pride/archive>, click on “log in” on the top right.

REVIEWERS' COMMENTS:

Reviewer #1 (Remarks to the Author):

The authors satisfactory responded to all my concerns.

Reviewer #2 (Remarks to the Author):

The authors have implemented all changes I had requested. I am very pleased with their answers and with the manuscript in its revised form.

W. Liebermeister

Reviewer #3 (Remarks to the Author):

At this point, the authors have thoroughly addressed my concerns, and seem to have addressed other reviewer concerns as well. I feel this is now an excellent study, and no other additional experiments or clarifications as necessary